# Primate TRIM5 proteins form hexagonal nets on HIV-1 capsids

Yen-Li Li[1†], Viswanathan Chandrasekaran[1†], Stephen D Carter[2], Cora L Woodward[2], Devin E Christensen[1], Kelly A Dryden[3], Owen Pornillos[3], Mark Yeager[3,4], Barbie K Ganser-Pornillos[3*], Grant J Jensen[2,5*], Wesley I Sundquist[1*]

[1]Department of Biochemistry, University of Utah, Salt Lake City, United States; [2]Division of Biology, California Institute of Technology, Pasadena, United States; [3]Department of Molecular Physiology and Biological Physics, University of Virginia School of Medicine, Charlottesville, United States; [4]Department of Medicine, Division of Cardiovascular Medicine, University of Virginia Health System, Charlottesville, United States; [5]Howard Hughes Medical Institute, California Institute of Technology, Pasadena, United States

**Abstract** TRIM5 proteins are restriction factors that block retroviral infections by binding viral capsids and preventing reverse transcription. Capsid recognition is mediated by C-terminal domains on TRIM5α (SPRY) or TRIMCyp (cyclophilin A), which interact weakly with capsids. Efficient capsid recognition also requires the conserved N-terminal tripartite motifs (TRIM), which mediate oligomerization and create avidity effects. To characterize how TRIM5 proteins recognize viral capsids, we developed methods for isolating native recombinant TRIM5 proteins and purifying stable HIV-1 capsids. Biochemical and EM analyses revealed that TRIM5 proteins assembled into hexagonal nets, both alone and on capsid surfaces. These nets comprised open hexameric rings, with the SPRY domains centered on the edges and the B-box and RING domains at the vertices. Thus, the principles of hexagonal TRIM5 assembly and capsid pattern recognition are conserved across primates, allowing TRIM5 assemblies to maintain the conformational plasticity necessary to recognize divergent and pleomorphic retroviral capsids.

**\*For correspondence:**
bpornillos@virginia.edu (BKG-P);
jensen@caltech.edu (GJJ); wes@
biochem.utah.edu (WIS)

[†]These authors contributed equally to this work

## Introduction

Mammalian hosts have evolved a series of different innate immune strategies to combat retroviruses (reviewed in [*Altfeld and Gale, 2015*; *Bieniasz, 2003*, *2004*; *Fitzgerald et al., 2014*; *Harris et al., 2012*; *Neil and Bieniasz, 2009*; *Rustagi and Gale, 2014*; *Sparrer and Gack, 2015*; *van Montfoort et al., 2014*; *Yoo et al., 2014*]). TRIM5α and the related TRIMCyp protein (collectively TRIM5) are restriction factors that recognize the capsid surfaces of incoming retroviral core particles, induce their dissociation, and inhibit reverse transcription (*Sayah et al., 2004*; *Stremlau et al., 2004*; *2006*). The mechanistic basis for core inactivation is not yet well established, but current models invoke the involvement of proteasomes (*Anderson et al., 2006*; *Campbell et al., 2008*; *Diaz-Griffero et al., 2007*; *Kutluay et al., 2013*; *Lukic et al., 2011*; *Rold and Aiken, 2008*; *Wu et al., 2006*), autophagosomes (*Mandell et al., 2014a*; *2014b*), and/or the establishment of a general anti-viral state (*Pertel et al., 2011*).

Like other members of the tripartite motif (TRIM) family (*Reymond et al., 2001*), TRIM5 proteins comprise a RING E3 ubiquitin (Ub) ligase domain (*Meroni and Diez-Roux, 2005*; *Yamauchi et al., 2008*), an L1 linker, a B-box 2 self-assembly domain (*Diaz-Griffero et al., 2009*; *Javanbakht et al.,*

**eLife digest**  After infecting a cell, a virus reproduces by forcing the cell to produce new copies of the virus, which then spread to other cells. However, cells have evolved ways to fight back against these infections. For example, many mammalian cells contain proteins called restriction factors that prevent the virus from multiplying. The TRIM5 proteins form one common set of restriction factors that act against a class of viruses called retroviruses.

HIV-1 and related retroviruses have a protein shell known as a capsid that surrounds the genetic material of the virus. The capsid contains several hundred repeating units, each of which consists of a hexagonal ring of six CA proteins. Although this basic pattern is maintained across different retroviruses, the overall shape of the capsids can vary considerably. For instance, HIV-1 capsids are shaped like a cone, but other retroviruses can form cylinders or spheres.

Soon after the retrovirus enters a mammalian cell, TRIM5 proteins bind to the capsid. This causes the capsid to be destroyed, which prevents viral replication. Previous research has shown that several TRIM5 proteins must link up with each other via a region of their structure called the B-box 2 domain in order to efficiently recognize capsids. How this assembly process occurs, and why it enables the TRIM5 proteins to recognize different capsids was not fully understood. Now, Li, Chandrasekaran et al. (and independently Wagner et al.) have investigated these questions.

Using biochemical analyses and electron microscopy, Li, Chandrasekaran et al. found that TRIM5 proteins can bind directly to the surface of HIV-1 capsids. Several TRIM5 proteins link together to form large hexagonal nets, in which the B-box domains of the proteins are found at the points where three TRIM5 proteins meet. This arrangement mimics the pattern present in the HIV-1 capsid, and just a few TRIM5 rings can cover most of the capsid.

Li, Chandrasekaran et al. then analysed TRIM5 proteins from several primates, including rhesus macaques, African green monkeys and chimpanzees. In all cases analyzed, the TRIM5 proteins assembled into hexagonal nets, although the individual units within the net did not have strictly regular shapes. These results suggest that TRIM5 proteins assemble a scaffold that can deform to match the pattern of the proteins in the capsid. Further work is now needed to understand how capsid recognition is linked to the processes that disable the virus.

2005; *Li and Sodroski, 2008*), an antiparallel dimeric coiled-coil, and an L2 linker that folds back on the coiled-coil (*Goldstone et al., 2014*; *Li et al., 2014*; *Sanchez et al., 2014*; *Weinert et al., 2015*) (*Figure 1A*). TRIM5 proteins also contain one of two different C-terminal viral core recognition domains, a B30.2/SPRY domain in TRIM5α (hereafter termed SPRY) or a cyclophilin A (CypA) domain in TRIMCyp (*Brennan et al., 2008*; *Newman et al., 2008*; *Nisole et al., 2004*; *Sayah et al., 2004*; *Stremlau et al., 2005*; *2006*; *Virgen et al., 2008*).

TRIM5 proteins act by binding the outer capsid shell of the viral core (*Biris et al., 2013*; *Black and Aiken, 2010*; *Diaz-Griffero et al., 2006b*; *Kar et al., 2008*; *Kovalskyy and Ivanov, 2014*; *Langelier et al., 2008*; *Sayah et al., 2004*; *Sebastian and Luban, 2005*; *Stremlau et al., 2006*; *Zhao et al., 2011*). The capsid protects and organizes the internal ribonucleocapsid, which comprises the viral NC protein, the RNA genome, and associated replicative enzymes. Closed retroviral capsids are constructed from several hundred CA protein hexamers and exactly 12 CA pentamers ([*Ganser et al., 1999*; *Gres et al., 2015*; *Li et al., 2000*; *Obal et al., 2015*; *Pornillos et al., 2011*; *Zhao et al., 2013*] and reviewed in [*Ganser-Pornillos et al., 2012*; *Zhang et al., 2015*]). Although all retroviral capsids are organized following these principles, individual capsids are unique, asymmetric objects that can differ in hexamer numbers and pentamer distributions (*Benjamin et al., 2005*; *Briggs et al., 2006*; *Butan et al., 2008*; *Ganser-Pornillos et al., 2004*; *Heymann et al., 2008*). For example, HIV-1 capsids are typically conical, but their sizes and cone angles can vary, and cylindrical and spherical capsids also form (*Benjamin et al., 2005*; *Briggs et al., 2006*; *Briggs et al., 2003*; *Ganser-Pornillos et al., 2004*; *Heymann et al., 2008*; *Welker et al., 2000*). Indeed, spherical and cylindrical capsids predominate in other retroviral genera, and capsid surface properties can vary considerably because CA proteins from different genera share low sequence identity (*Berthet-Colominas et al., 1999*; *Campos-Olivas et al., 2000*; *Cornilescu et al.,*

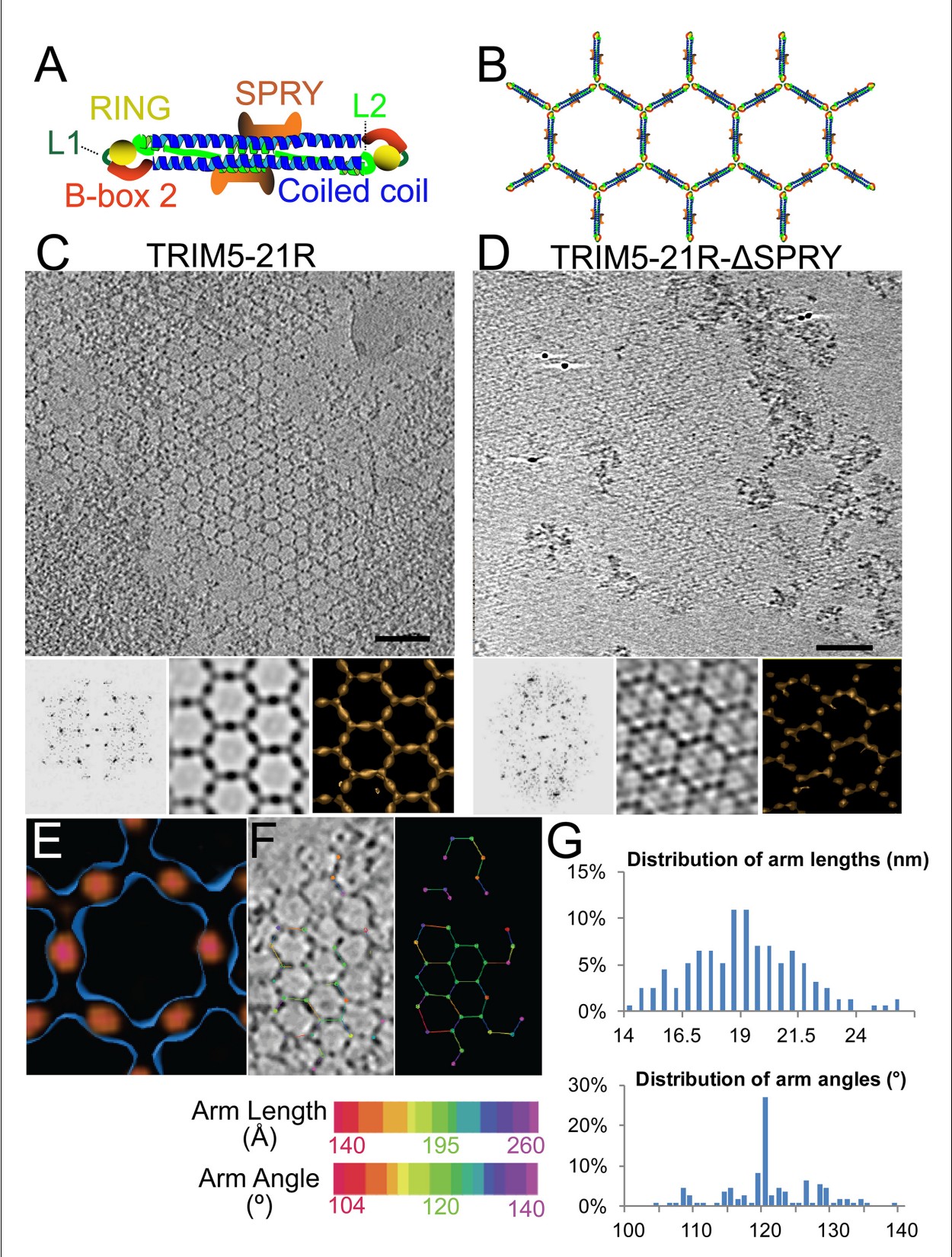

**Figure 1.** ECT analysis of TRIM5-21R 2D crystals. (A) Schematic of the TRIM5 dimer. The two RING (yellow) and B-box 2 (red) domains are separated by a ~17 nm, antiparallel dimeric coiled-coil (blue). The two L2 linkers (green) fold back towards the 2-fold axis of the coiled-coil to orient two capsid-

*Figure 1 continued on next page*

*Figure 1 continued*

binding SPRY domains (orange). (B) Schematic of the TRIM5 hexagonal lattice model. (C and D) Tomographic slice (top) of (C) full-length TRIM5-21R, and (D) TRIM5-21R$_{\Delta SPRY}$ lattices. Scale bars are 100 nm. In both cases, the computed Fourier transform (bottom, left) and subtomogram average without imposed rotational symmetry (bottom, middle) exhibit six-fold symmetry. Iso-surface representations of the densities are also shown (bottom, right). (E) A density difference map of the subtomogram averages of full-length TRIM5-21R and TRIM5-21R$_{\Delta SPRY}$ lattices reveals positive density (red) at the center of each hexagon edge, corresponding to the SPRY domain position, supporting the TRIM5α dimer and hexamer models shown in (A and B). (F) Heat maps (bottom) of lattice arm lengths and angles measured from refined lattice points (top) selected from the TRIM5-21R tomogram in (C). (G) Histograms showing the distributions of measured arm lengths (n = 155) and angles (n = 111). The most abundant arm length (18.5–19 nm) and arm angle (120°) are consistent with the structure models in (A and B) and the p6 plane group symmetry of TRIM5-21R 2D crystals (*Ganser-Pornillos et al., 2011*). Note that this analysis probably underestimates the extent of hexamer variability owing to the initial selection of well-ordered lattice points.

*2001*; *Ganser et al., 1999*; *Jin et al., 1999*; *Khorasanizadeh et al., 1999*; *Kingston et al., 2000*; *Momany et al., 1996*; *Mortuza et al., 2008*; *Mortuza et al., 2004*; *von Schwedler et al., 1998*; *Zlotnick et al., 1998*).

To function effectively, individual TRIM5 proteins must overcome these variations in retroviral capsid shape and sequence (*Hatziioannou et al., 2003*; *Wilson et al., 2008*). We, and others, have proposed that TRIM5 proteins recognize pleomorphic capsids by recognizing repeating patterns on the capsid surface (*Biris et al., 2012*; *Ganser-Pornillos et al., 2011*; *Goldstone et al., 2014*; *Yang et al., 2012*). This model supposes that flexible loops on the SPRY and CypA domains can adopt multiple different conformations and can bind weakly to conserved elements on the capsid surfaces (*Biris et al., 2012*; *Caines et al., 2012*; *Kovalskyy and Ivanov, 2014*; *Price et al., 2009*; *Song et al., 2005a*; *Stremlau et al., 2005*; *Ylinen et al., 2010*). These weak interactions are then amplified by TRIM5 assembly into a higher-order hexagonal lattice, which positions arrays of SPRY/CypA domains to interact with repeating epitopes on the capsid surfaces (*Ganser-Pornillos et al., 2011*; *Li and Sodroski, 2008*).

This 'pattern recognition' model has been supported by biochemical and structural analyses of a TRIM5 protein construct called TRIM5-21R, which is an artificial chimera in which the RING domain from human TRIM21 replaces the RING domain of rhesus TRIM5α (*Diaz-Griffero et al., 2006a*; *Kar et al., 2008*; *Langelier et al., 2008*). The TRIM5-21R construct retains HIV-1 restriction activity, and has been used in several studies owing to its unusually favorable stability, solubility and assembly properties. Consistent with the pattern recognition model, TRIM5-21R was shown to assemble into open hexagonal lattices, both alone and on the surface of 2D CA crystals that mimic the surface of the HIV-1 capsid (*Ganser-Pornillos et al., 2011*). TRIM5-21R assemblies could be microns in size but lacked strict long-range order, and 2D projections could therefore only be reconstructed to a resolution of ~7.5 nm. Domain positions therefore had to be inferred, and were interpreted in the absence of high-resolution information on the structure of the TRIM5 protein core.

Technical challenges in purifying authentic HIV-1 cores have also been a significant experimental limitation, and all published biochemical and structural studies of TRIM5α-capsid interactions have therefore either employed crude viral core preparations or artificial mimics of the capsid surface (*Black and Aiken, 2010*; *Ganser-Pornillos et al., 2011*; *Langelier et al., 2008*; *Sebastian and Luban, 2005*; *Stremlau et al., 2006*; *Zhao et al., 2011*). Thus, the interactions between authentic viral capsids and TRIM5 proteins have yet to be investigated biochemically or structurally. To address these different shortcomings, we have developed methods for preparing authentic recombinant TRIM5 proteins, co-assemblies of TRIM5 and CA proteins, and stable HIV-1 cores. These reagents were then used to demonstrate that TRIM5 proteins form hexagonal arrays on HIV-1 capsids.

## Results

### Structure-based models for TRIM5-21R assembly

A composite model for the domain organization within the TRIM5α dimer is shown in *Figure 1A*. We (*Sanchez et al., 2014*), and others (*Goldstone et al., 2014*; *Weinert et al., 2015*), have suggested that these dimers form the edges of the hexameric rings observed within the hexagonal TRIM5-21R lattice (modeled in *Figure 1B*). This model is attractive because the edges of each

hexagon are ~19 nm long, which would accommodate the antiparallel TRIM5α coiled-coil (~17 nm), and the RING and B-box 2 domains would associate at the three-fold vertices, which is consistent with the trimeric B-box 2 domain structure presented in the accompanying paper (*Wagner et al., 2016*). Finally, the L2 linkers fold back and form a four-helix bundle at the center of the coiled-coil, and this platform could, in principle, buttress and orient the TRIM5α SPRY domains. None of the domain positions have been determined experimentally, however, and we therefore began our studies by defining the location of the SPRY domains within the hexagonal TRIM5-21R lattice.

To visualize TRIM5-21R assemblies in three dimensions and define the SPRY domain positions, we generated electron cryotomograms (ECT) from tilt series of vitrified 2D crystals of both full length TRIM5-21R (*Figure 1C*) and a construct that lacked the SPRY domain (TRIM5-21R$_{\Delta SPRY}$, residues 1–300, *Figure 1D*). The 3D reconstructions were refined and improved by subtomogram averaging of densities centered at equivalent lattice vertices. As expected, TRIM5-21R and TRIM5-21R$_{\Delta SPRY}$ both assembled into similar planar lattices of hexagonal rings, with inter-ring spacings and protein densities matching those of the previous 2D projection structures (*Ganser-Pornillos et al., 2011*). Difference density maps clearly revealed that the SPRY domains are located at the center of each hexagon edge (*Figure 1E*), thereby supporting the model shown in *Figure 1B*.

## The TRIM5-21R lattice is a hexagonal net with variable arm lengths and angles

Although the paracrystalline arrays of TRIM5-21R exhibited long-range order, they diffracted poorly, suggesting variability within the lattice. To quantify this variability, we performed a nearest neighbor analysis of the refined positions of the lattice vertices used for subtomogram averaging. The relative positions of 153 vertices in the TRIM5-21R 2D crystals were used to define individual hexamer edge lengths and angles (see *Figure 1F*). The length distribution of hexamer edges was centered about a mean of 19 nm, but individual lengths varied by up to ±5 nm (*Figure 1G*, upper panel). Similarly, the distribution of hexamer vertex angles was centered about 120°, but varied by up to ±20° (*Figure 1G*, lower panel). Thus, individual rings within the TRIM5 lattice exhibited considerable conformational variability, explaining the poor crystalline order and modest diffraction resolution.

## Expression and purification of authentic primate TRIM5α and TRIMCyp proteins

Recombinant owl monkey TRIMCyp has been purified (*Pertel et al., 2011*), but TRIM5α proteins are more challenging to purify because these proteins tend to self-assemble, both in cells and in vitro. Hence, previous biochemical and structural studies of TRIM5α proteins have been performed with impure proteins, protein fragments, or non-native chimeric constructs (*Biris et al., 2012*; *Goldstone et al., 2014*; *Kar et al., 2008*; *Langelier et al., 2008*; *Sanchez et al., 2014*; *Yang et al., 2012*; *Ganser-Pornillos et al., 2011*). To overcome this limitation, we tested a variety of different expression and purification conditions, with the goal of developing a general method for preparing milligram quantities of authentic, full-length primate TRIM5α and TRIMCyp proteins.

The strategy that was ultimately successful entailed expressing TRIM5 proteins in insect cells using a baculoviral expression system. As described in greater detail in the Materials and methods, expressed TRIM5 proteins formed cytoplasmic bodies that could be solubilized by lysing the cells in a low ionic strength, alkali buffer that contained the non-ionic detergent Triton X-100, as well as a non-detergent small molecule, sulfobetaine-256 (NDSB-256) that has previously been shown to inhibit protein aggregation (*Sainsbury et al., 2008*; *Vuillard et al., 1995*). Once solubilized, primate TRIM5 proteins typically remained dimeric and soluble under low salt, alkaline conditions in the absence of Triton X-100 and NDSB-256, even at concentrations greater than 1 mg/ml. The proteins could therefore be purified, provided they were maintained at high pH, low salt and/or low protein concentrations.

Our stepwise protein purification protocol is illustrated for rhesus TRIM5α in *Figure 2A*. Briefly, various N-terminal OneSTrEP-FLAG- (OSF-) or C-terminal FLAG-OneSTrEP- (FOS-) tagged TRIM5 proteins were initially purified using Strep-Tactin affinity chromatography, the affinity tag was removed by HRV14-3C protease treatment, and the proteins were then purified to homogeneity by anion exchange and gel filtration chromatography. Analogous approaches were used to express and purify wild type and mutant TRIM5α proteins from rhesus macaques (*Macaca mulatta*, here

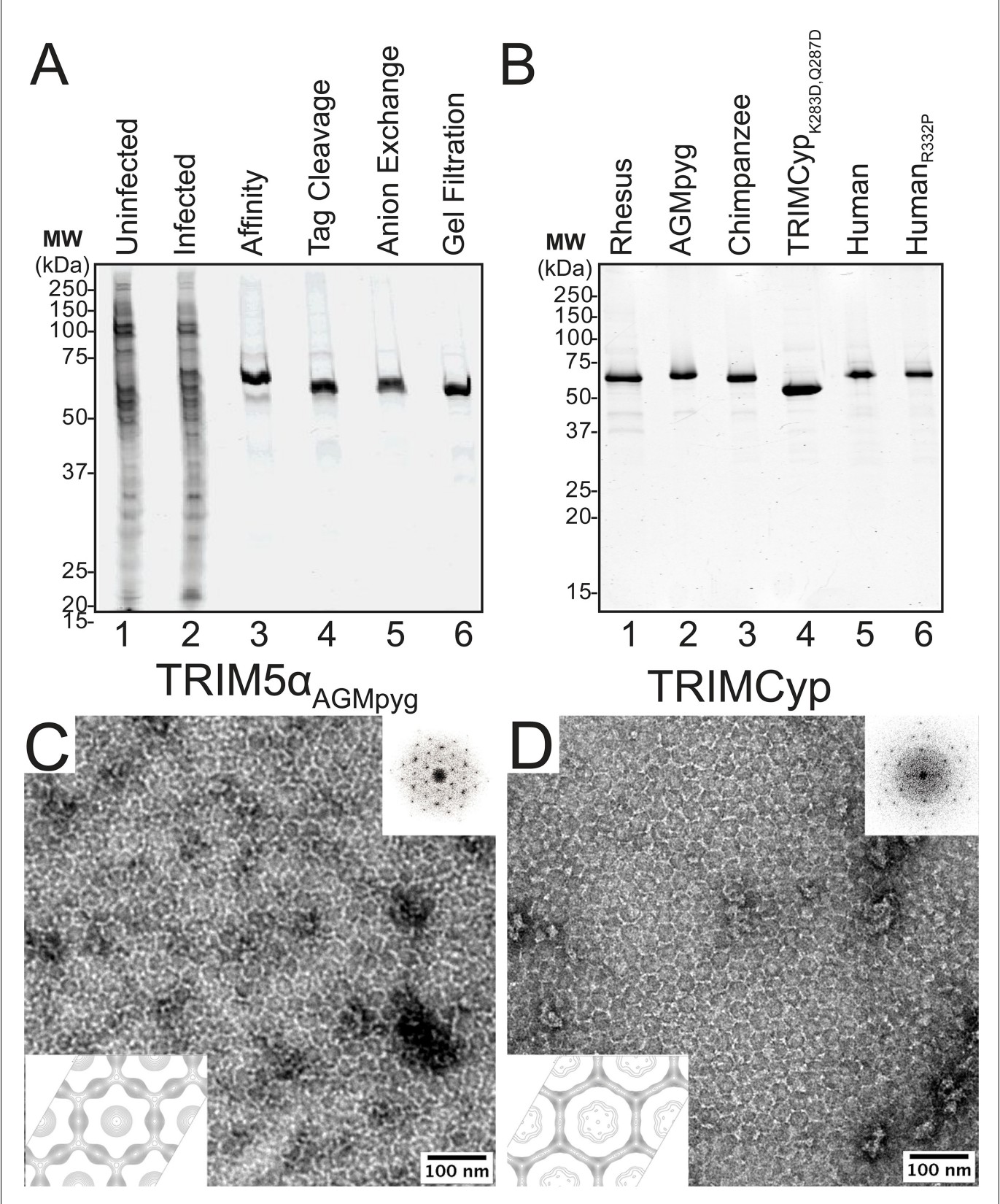

**Figure 2.** Purification and characterization of recombinant TRIM5 proteins. (**A**) Coomassie-stained SDS-PAGE showing the stepwise purification of rhesus TRIM5α (TRIM5α_{rh}). Samples correspond to: soluble lysate from control SF9 cells (Uninfected, lane 1); soluble lysate from SF9 cells expressing

*Figure 2 continued on next page*

*Figure 2 continued*

OSF-TRIM5α$_{rh}$ (Infected, lane 2); Strep-Tactin affinity-purified OSF-TRIM5α$_{rh}$ (Affinity, lane 3); TRIM5α$_{rh}$ after removal of the OSF tag by HRV14-3C protease treatment (Tag Cleavage, lane 4); dimeric TRIM5α$_{rh}$ purified by Q anion exchange chromatography (Anion Exchange, lane 5); dimeric TRIM5α$_{rh}$ purified by Superdex 200 gel filtration chromatography (Gel Filtration, lane 6). (B) Coomassie-stained SDS-PAGE showing 1.5 µg of purified rhesus, African green monkey pygerythrus (AGMpyg), chimpanzee TRIM5α, proteolysis-resistant owl monkey TRIMCyp$_{K283D,Q287D}$, human TRIM5α, and HIV-1-restricting human TRIM5α$_{R332P}$. (C,D) TRIM5 hexagonal assembly is a conserved property of primate TRIM5 proteins. Negatively stained EM image of hexagonal arrays formed by (C) TRIM5α$_{AGMpyg}$ and (D) TRIMCyp. Computed Fourier transforms (top right insets) show clear hexagonal order and filtered projection density maps of 2-dimensional crystals (bottom left insets) also reveal hexagonal rings and density distributions reminiscent of TRIM5-21R lattices (*Ganser-Pornillos et al., 2011*). The unit cell parameters are a = 345 Å, b = 345 Å, γ = 120° (TRIM5α$_{AGMpyg}$); and a = 345 Å, b = 344 Å, γ = 119° (TRIMCyp). Note that the TRIMCyp samples contained a mixture of full-length TRIMCyp and fragments that were proteolyzed to the C-terminus of residues K283 or Q287 (see Results and Materials and methods for details). The relatively thinner two fold density in the TRIMCyp projection map could either reflect low crystal occupancy of the CypA domain (due to proteolysis) or inherently flexible CypA domains in TRIMCyp as has been proposed by (*Goldstone et al., 2014*).

The following figure supplement is available for figure 2:

**Figure supplement 1.** HIV-1 CA restriction activity of different TRIM5 alleles.

abbreviated TRIM5α$_{rh}$), African green monkeys (*Chlorocebus pygerythrus*, TRIM5α$_{AGMpyg}$), chimpanzees (*Pan troglodytes*, TRIM5α$_{cpz}$), humans (*Homo sapiens*, TRIM5α$_{hu}$), and the TRIMCyp protein from owl monkeys (*Aotus trivirgatus*, TRIMCyp). Yields ranged between 1.3 and 9.6 mg/L of insect cell cultures, and all of the proteins eluted with similar retention times during the final gel filtration chromatography step, indicating that they were all dimers of similar shape. All of the proteins could be purified to >95% purity (*Figure 2B*) with the exception of TRIMCyp, where our preparations also contained breakdown contaminants that mapped to proteolytic cleavage at residues Lys283 and Gln287 (not shown). These breakdown contaminants were eliminated by creating a mutant construct that expressed TRIMCyp$_{K283D,Q287D}$ (*Figure 2B*, lane 4). These mutations are not expected to affect functionally relevant properties of the protein because TRIMCyp$_{K283D,Q287D}$ retains potent HIV-1 restriction activity (*Figure 2—figure supplement 1*).

## Conservation of hexagonal TRIM5 protein assembly

To determine whether the ability to assemble into hexagonal nets is a conserved property of authentic TRIM5 proteins, we screened for conditions that promoted assembly of different primate TRIM5 proteins, using negative stain EM imaging to assay assembly states. These screens identified conditions under which two of the TRIM5 proteins, TRIM5α$_{AGMpyg}$ (*Figure 2C*) and TRIMCyp (*Figure 2D*) spontaneously formed 2D hexagonal assemblies that were similar in appearance to those formed by TRIM5-21R (*Ganser-Pornillos et al., 2011*). Assembly efficiencies varied, however, as TRIM5α$_{AGMpyg}$ assembled very efficiently under the same conditions as TRIM5-21R, whereas TRIMCyp assembled inefficiently and required additional precipitants (see Materials and methods). Negatively stained 2D crystals of TRIM5α$_{AGMpyg}$ and TRIMCyp were imaged and processed to generate Fourier-filtered 2D projection reconstructions (*Figure 2C,D*; bottom). Both TRIM5 proteins formed lattices comprising open hexameric rings that were similar in appearance and size to the TRIM5-21R rings, demonstrating that diverse TRIM5 proteins from different primates that share 73% pairwise identity across their TRIM domains share the ability to assemble into analogous hexagonal nets.

## Templated hexagonal TRIM5 protein assembly on HIV-1 CA surfaces

The pattern recognition model for TRIM5 restriction predicts that binding to the surface of the viral capsid promotes hexagonal TRIM5 assembly. We therefore tested whether hexagonal 2D crystals of HIV-1 CA, which mimic the capsid surface, could promote the assembly of three different restricting TRIM5 proteins; TRIM5α$_{rh}$, TRIMCyp$_{K283D,Q287D}$, and TRIM5α$_{hu,R332P}$, and two different non-restricting TRIM5 proteins; wild type TRIM5α$_{hu}$ and TRIM5α$_{cpz}$ (*Hatziioannou et al., 2004*; *Nisole et al., 2004*; *Sayah et al., 2004*; *Song et al., 2005b*; *Stremlau et al., 2004*; *2005*; *Yap et al., 2005*). To test for templated assembly of TRIM5 proteins, soluble dimeric proteins were incubated together with preassembled 2D CA crystals under solution conditions that were sufficiently stringent to prevent untemplated assembly. TRIM5α$_{AGMpyg}$ was not used in these studies because it assembled very robustly even in the absence of a template.

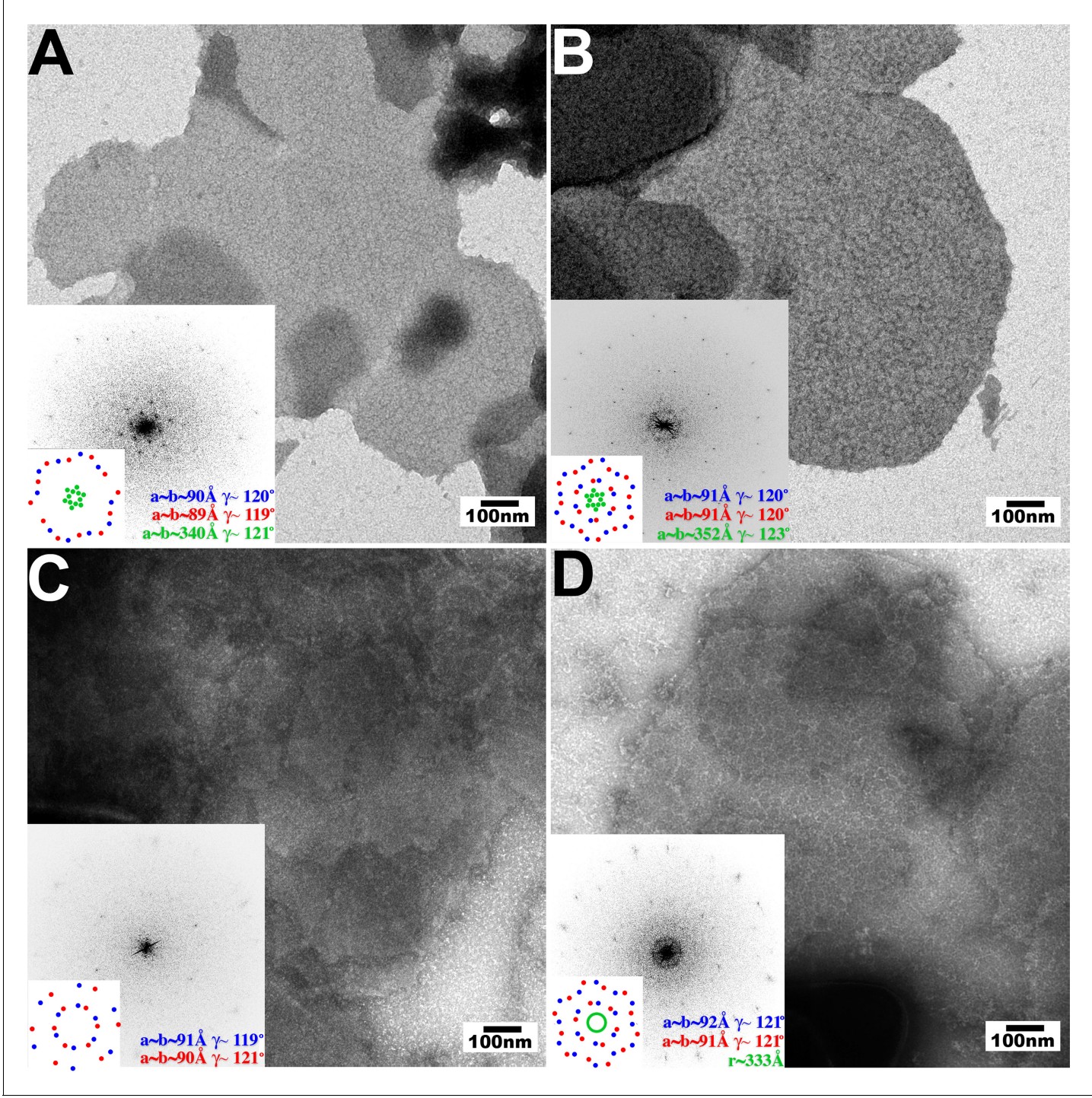

**Figure 3.** Assembly of restricting TRIM5 proteins on 2D crystals of HIV-1 CA. Negative stain EM images of CA 2D crystals decorated with (**A**) TRIM5α$_{rh}$, (**B**) TRIMCyp$_{K283D,Q287D}$, (**C**) TRIM5α$_{hu}$ (non-restricting allele), (**D**) TRIM5α$_{hu,R332P}$ (restricting mutant). Scale bars are 100 nm. Computed Fourier transforms (insets) and indexing (second insets) show the first and second order reflections of two CA lattices and their unit cell parameters (red and blue) as well as diffraction spots (**A**, **B**) or rings (**D**) corresponding to the first order reflections of the TRIM5 lattices (green).

As shown in *Figure 3*, the three restricting TRIM5 proteins assembled into visible hexagonal nets on the surfaces of preformed HIV-1 CA crystals, whereas templated assembly was not observed for either one of the non-restricting TRIM5 proteins (*Figure 3C* and data not shown). Templated

assembly therefore correlated well with restriction activity, emphasizing the coupling of CA binding and TRIM5 lattice assembly. Computed Fourier transforms of the images of decorated crystals (*Figure 3*, insets) revealed well-defined first- and second-order reflections from the smaller underlying CA lattice (red and blue), as well as more diffuse peaks (TRIM5α$_{rh}$ and TRIMCyp$_{K283D,Q287D}$, green) or a powder diffraction ring (TRIM5α$_{hu,R332P}$) corresponding to the first-order reflections from the hexagonal TRIM5 lattices. Thus, all three restricting TRIM5 proteins bound the CA surfaces and assembled into hexagonal nets that were clearly visible, but lacked extensive crystalline order.

## TRIM5 binding to helical HIV-1 CA tubes

The helical tubes formed by pure recombinant HIV-1 CA provide another regularized model for the curved, symmetric arrays of CA hexagons on conical viral capsid surfaces (*Campbell and Vogt, 1995*; *Li et al., 2000*). We employed a sucrose co-sedimentation assay to test whether the restricting TRIM5α$_{AGMpyg}$ and TRIMCyp$_{K283D,Q287D}$ proteins bound disulfide-crosslinked helical tubes formed by a mutant HIV-1 CA protein that assembled into discrete helical tubes stabilized by intra-hexamer disulfide crosslinks (CA$_{A14C,E45C,A92E}$) (*Byeon et al., 2009*; *Ganser-Pornillos et al., 2011*; *Langelier et al., 2008*; *Li et al., 2000*; *Pornillos et al., 2009*; *2010*; *Zhao et al., 2011*). Both TRIM5α$_{AGMpyg}$ and TRIMCyp$_{K283D,Q287D}$ bound the CA tubes, as judged by their co-sedimentation through the sucrose cushion when the CA tubes were present, but not when they were absent (compare the 'pellet' fractions in lanes 1 vs. 2 and lanes 5 vs. 6 in *Figure 4—figure supplement 1A*). Binding was specific because TRIM5α$_{AGMpyg}$ did not co-sediment with CA tubes when the SPRY domain was removed (compare lanes 2 vs. 4) and because TRIMCyp$_{K283D,Q287D}$ did not co-sediment with CA tubes in the presence of cyclosporine A, which competitively inhibits the CypA-CA interaction (compare lanes 6 vs. 7). Thus, pure recombinant TRIM5 proteins bind the hexagonal lattices of both helical CA tubes and 2D CA crystals.

Negative stain and deep-etch electron microscopy were used to image the TRIM5-decorated HIV-1 CA tubes. Four restricting TRIM5 proteins, TRIM5α$_{AGMpyg}$, TRIMCyp$_{K283D,Q287D}$, TRIM5α$_{rh}$, and TRIM5α$_{hu,R332P}$ formed thin ring-like decorations on the surfaces of CA tubes (*Figure 4—figure supplement 1B*). The TRIM5 decorations typically appeared as light, string-like nets against the darker underlying CA tubes when the assemblies were stained with either uranyl acetate (UA) or phosphotungstate (PTA). Equivalent decorations were not observed for control CA tubes alone or for CA tubes plus TRIM5α$_{hu}$, which does not restrict HIV-1. TRIM5 decoration of CA tubes therefore correlated with restriction activity.

Deep-etch EM images often exhibit even greater contrast than negative stain transmission EM images, and this effect was evident in deep-etch images of undecorated HIV-1 CA tubes, where rows of individual CA hexamers were readily visible (*Figure 4A*, upper panel). Strings and rings of TRIM5α$_{AGMpyg}$ were often readily visible on the decorated CA tube surfaces, and networks of rings were sometimes observable (*Figure 4A*, lower panel). Hence, TRIM5 proteins can form hexagonal nets on the surfaces of helical HIV-1 CA tubes.

The efficiency of TRIM5 assembly on disulfide-crosslinked CA tubes was relatively low, which may explain why the decorations were not noted in previous studies. We therefore screened additional assembly conditions to determine the source of this variability. These experiments revealed that co-incubation of native CA protein with TRIM5α under low ionic strength conditions produced CA tubes that were extensively decorated with TRIM5α. Co-assembly was reproducible, and both tagged and untagged TRIM5-21R, TRIM5$_{AGMpyg}$, and TRIM5$_{rh}$ proteins formed ring-like decorations on CA tube surfaces at a variety of different CA/TRIM5 molar ratios (*Figure 4B*, data not shown, and *Wagner et al., 2016*). The TRIM5α$_{AGMpyg}$ inter-ring spacing distribution on the tubes centered at 30–35 nm, as was also the case for TRIM5α$_{AGMpyg}$ assemblies on pre-formed disulfide-crosslinked CA tubes (*Figure 4C*). In all cases, the rings were irregular and their spacings varied between 15 and 55 nm. Electron cryotomography (ECT) analyses similarly revealed extensive networks of hexagonal nets coating individual cylinders (*Figure 4D* and *Video 1*). As seen for TRIM5-21R 2D crystals, the hexamer edge lengths measured from three-fold vertices selected from the tomogram were predominantly ~19 nm, but varied between 15 to 23 nm. Similarly, the hexamer vertex angles averaged ~120°, but ranged from 80° to 160° (*Figure 4E*). The flexibility of the TRIM5 assembly may reflect flexing of the coiled-coil domains and/or the hinge between B-box 2 and coiled-coil domains (*Wagner et al., 2016*). In summary, TRIM5 proteins consistently formed hexagonal arrays on curved HIV-1 CA lattices, and the TRIM5 lattice was more extensive when the two proteins co-assembled

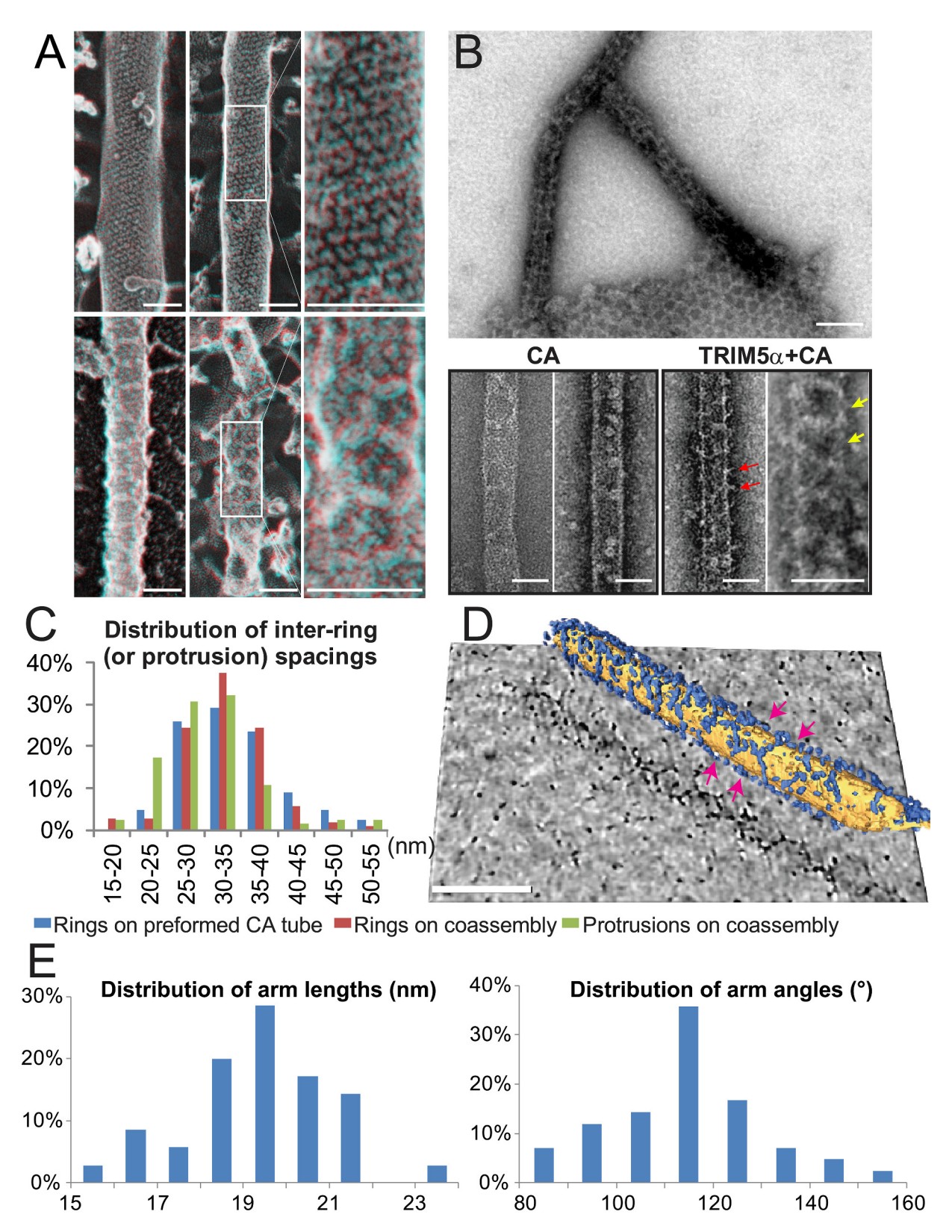

**Figure 4.** Assembly of TRIM5α proteins on HIV-1 CA tubes. (**A**) Deep-etch electron micrographs of control hyperstable CA tubes (top) and TRIM5α decorated CA tubes (bottom) with blow-up views of boxed regions to the right. Scale bar is 50 nm. (**B**) Negative stain electron micrographs of co-

*Figure 4 continued on next page*

*Figure 4 continued*

assembled TRIM5α_AGMpyg-coated CA tubes (top). Scale bar is 100 nm. Expanded views of negatively stained CA assembly in the absence (bottom left) or presence of TRIM5α_AGMpyg proteins (bottom right). TRIM5α formed ring-like decorations (yellow arrows) on the tube surface and displayed protrusions along the edge of the tube (red arrows). Similar decorations were not observed in the control case. The protrusions were regularly spaced and we speculate that they are either well-ordered coiled-coil arms from adjacent hexagons wrapping around the CA tubes or possibly ordered RING domains projecting outward from the lattice. Scale bars are 50 nm. (C) Histograms showing the distributions of measured inter-ring spacings in the TRIM5α_AGMpyg–decorated tubes (blue bars, n = 170) and in co-assemblies (red bars, n = 164), and of inter-protrusion spacings in co-assemblies (green bars, n = 166). The most abundant inter-ring spacing (30–35 nm) is consistent with the spacing of TRIM5-21R 2D crystals, indicating similar structures. (D) Electron cryotomography (ECT) reveals that TRIM5α forms hexagonal nets on the surface of CA tube. Hexagon-like rings are marked by magenta arrows. Scale bar is 80 nm. (E) Histograms showing the distributions of arm lengths (n = 35) and angles (n = 42) measured from three-fold vertices in ECT.

The following figure supplement is available for figure 4:

**Figure supplement 1.** TRIM5 protein binding to hyperstable HIV-1 CA tubes.

together, presumably because the two lattices could template one another and thereby optimize their interactions.

## Generation of hyperstable, disulfide-crosslinked HIV-1 core particles

A central goal of our studies was to analyze TRIM5 binding to authentic viral capsids, both biochemically and by direct imaging. This goal is technically challenging owing to the inherent instability of viral core particles. We reasoned that this challenge might be overcome by disulfide-crosslinking the capsid shell to increase its stability. This strategy was particularly attractive because previous studies showed that Cys residues substituted at CA positions Ala14 and Glu45 crosslinked efficiently when CA hexamers were assembled in vitro (*Pornillos et al., 2009*; *2010*). We therefore tested the effect of introducing these substitutions into authentic HIV-1 capsids.

Normal levels of viral particles were produced from 293T cells that expressed an HIV-1_NL4-3ΔR8.2 proviral expression vector that encoded the mutant $CA_{A14C,E45C}$ protein (not shown). Wild type and mutant viral cores were isolated by centrifugation of virions through a detergent layer to remove the outer viral membrane and then directly into a 30–70% sucrose gradient, where viral cores concentrated at a density of 1.22–1.27 g/ml (fractions 10–12, highlighted in pink in *Figure 5A*). As reported previously (*Forshey et al., 2002*; *Kotov et al., 1999*), wild type cores could also be isolated using this procedure (see *Figure 5B*). However, recovered core yields were consistently modest in our hands (0.2 ± 0.1% based upon total virion CA), apparently because most of the CA molecules dissociated from the core and migrated to the top of the gradient during purification (*Figure 5A*, left panel). In contrast, nearly all of the $CA_{A14C,E45C}$ protein migrated toward the bottom of the gradient when cores were isolated from mutant virions (*Figure 5A*, right panel). Much of this CA protein was present within small incomplete, broken, or spherical assemblies that concentrated at densities of 1.18–1.21 g/ml (gradient fractions 6–9, *Figure 5A*, right panel and see *Figure 5—figure supplement 1*). These non-native assemblies probably arose from spurious crosslinking of CA hexamers in the free intraviral pool of CA molecules that are excluded from the mature HIV-1 capsid (*Benjamin et al., 2005*; *Briggs et al., 2004*; *Lanman et al., 2004*; *Monroe et al., 2010*). Nevertheless, a substantial fraction of the CA protein also concentrated at the density expected for native core particles (fractions 10–12, highlighted in pink in *Figure 5A*, right panel). These 'hyperstable' cores were reproducibly recovered in higher yields (0.8 ± 1%) than wild type cores (0.2 ± 0.1%), and their morphologies were similar to wild type cores (*Figure 5B*, right panel). Consistent with the design, nearly

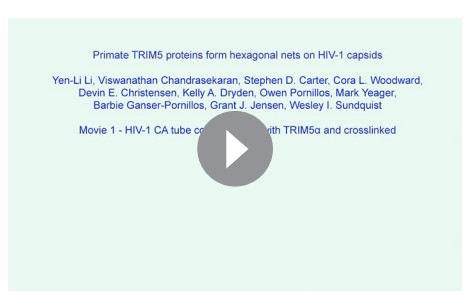

**Video 1.** HIV-1 CA tube co-assembled with TRIM5α and crosslinked. ECT of HIV-1 CA tube (yellow) decorated with TRIM5α (blue). Video corresponds to *Figure 4D*.

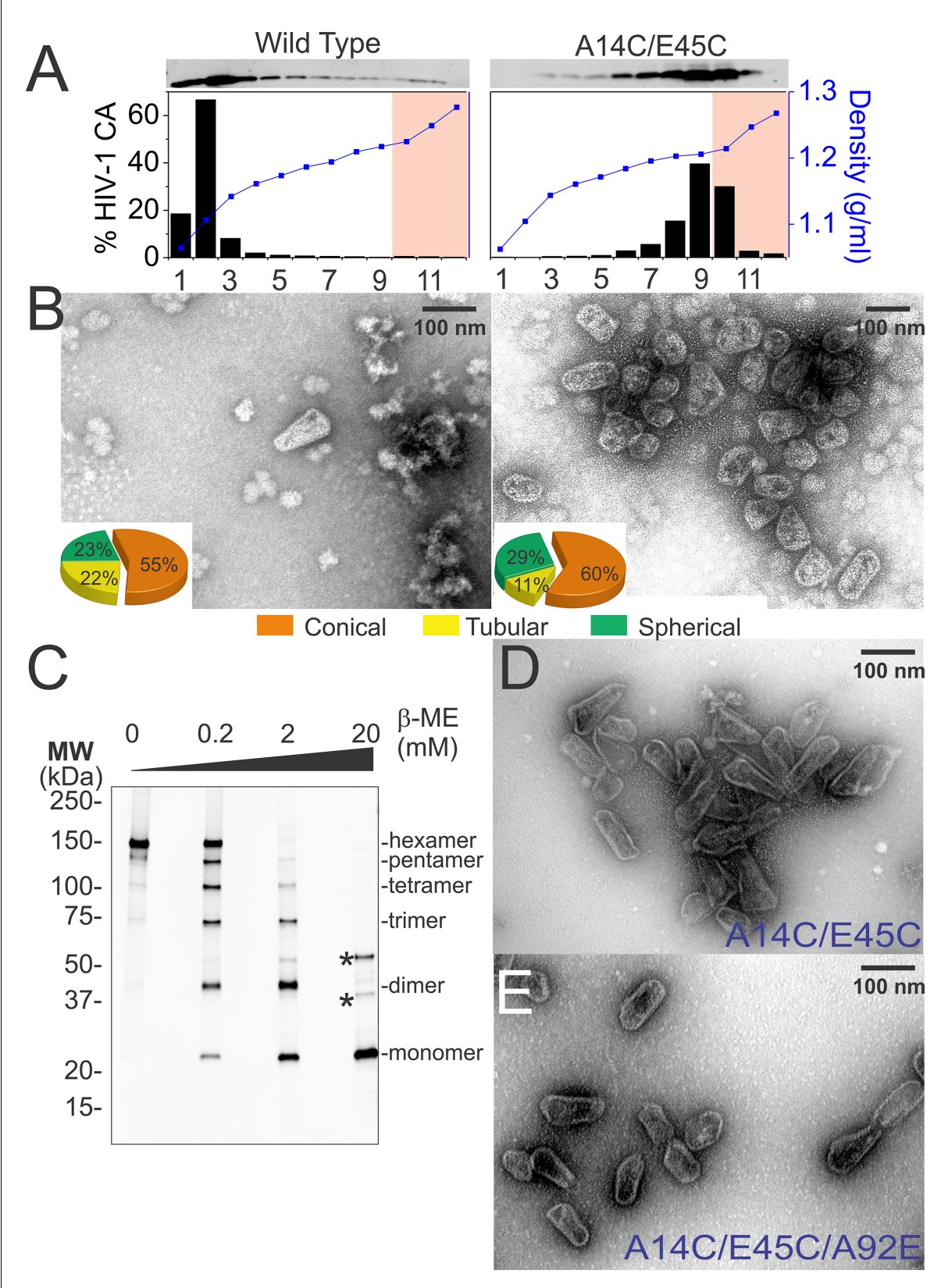

**Figure 5.** Purification of wild type and hyperstable HIV-1 cores. (**A**) Sucrose-gradient purification profiles of wild type (left) and hyperstable A14C/E45C (right) HIV-1 cores (**Kotov et al., 1999**). (top) α-CA western blots of sucrose gradient fractions, (bottom) graph showing quantified CA levels
*Figure 5 continued on next page*

*Figure 5 continued*

(histogram) and solution density (blue line; g/ml) in each gradient fraction. The higher stability of crosslinked HIV-1 cores can be seen by comparing the amounts of wild type and CA$_{A14C/E45C}$ in fractions 10–12 (pink regions). Core-containing fractions were pooled, washed, and concentrated for the analyses in B and C. The experiment was repeated at least three times with similar results. (B) Negative stain electron micrographs of wild type (left) and CA$_{A14C/E45C}$ (right) cores. Pie charts (inset) of observed morphologies of wild type (left; n = 143 cores) and CA$_{A14C/E45C}$ cores (right; n = 353 cores) reveal that the introduced Cys crosslinks do not alter HIV-1 core morphologies significantly. (C) Non-reducing α-CA western blots showing that CA$_{A14C/E45C}$ cores are crosslinked (compare % hexamer as a function of [β-ME]). Asterisks indicate unprocessed Gag fragments (p55 and p41) that co-purified with mature cores during purification. (D, E) Negative stain electron micrographs showing the relative abundance and purity of cores purified by the affinity method. (D) CA$_{A14C/E45C}$ crosslinked cores and (E) CA$_{A14C/E45C/A92E}$ crosslinked cores. Note that the additional A92E mutation reduced core clustering. Scale bar is 100 nm.

The following figure supplement is available for figure 5:

**Figure supplement 1.** Hyperstable core particle characterization and purification.

all of the CA molecules within these fractions were crosslinked within stable hexamers, as analyzed by non-reducing SDS-PAGE and western blotting (*Figure 5C*). These experiments indicate that disulfide crosslinking occurs spontaneously in otherwise native and untreated HIV-1 capsids, and that the crosslinks stabilize the capsids without introducing any major morphological defects.

## Isolation of HIV-1 core particles for cryoEM imaging

Hyperstable HIV-1 cores purified on sucrose gradients were not optimal for imaging studies because they aggregated and co-sedimented with vesicles and other impurities. To produce purer cores, we designed an alternative core affinity purification method that exploited the interaction between cyclophilin A (CypA) and HIV-1 CA (outlined in *Figure 5—figure supplement 1D*) (*Franke et al., 1994*; *Gamble et al., 1996*; *Luban et al., 1993*; *Thali et al., 1994*). Viral membranes were stripped by a brief Triton X-100 treatment and the liberated cores were then captured on magnetic Strep-Tactin beads derivatized with OSF-CypA. The immobilized cores were washed rigorously and then eluted with the small molecule cyclosporine A (CsA), which competitively inhibits the CypA-CA interaction and binds CypA ~700-fold more tightly than does CA (*Figure 5D* and *Figure 5—figure supplement 1E*) (*Braaten et al., 1996*; *Franke and Luban, 1996*; *Gamble et al., 1996*; *Wear et al., 2005*; *Yoo et al., 1997*). This method increased core yields by an additional four fold (3 ± 2% core recovery) and increased their purity (compare *Figure 5D* to *Figure 5B*, right panel). The method also reduced the fraction of broken cores, possibly because they were less stable and therefore removed during the extensive wash steps and/or because they bound less avidly to the matrix. To reduce core clustering, we introduced the CA A92E substitution within the exposed loop of HIV-1 CA. This mutation does not affect TRIM5α restriction (*Li et al., 2006*), and was previously shown to reduce the clustering of helical CA tubes, presumably by reducing overall surface hydrophobicity (*Ganser-Pornillos et al., 2004*; *Li et al., 2000*; *Zhao et al., 2011*). As shown in *Figure 5E*, the substitution also reduced the clustering of viral cores (compare *Figures 5D and E*), and did so without altering core morphology or reducing core yields (3 ± 1% core recovery).

In summary, dispersed hyperstable CA$_{A14C,E45C,A92E}$ cores could be purified by affinity chromatography in high yields. The purified cores contained the expected viral proteins as analyzed by SDS-PAGE with silver staining (*Figure 5—figure supplement 1F*), were hyperstable and fully disulfide crosslinked (*Figure 5C*), exhibited normal capsid morphologies (compare *Figure 5E* to *Figure 5B*), and spread diffusely on EM grids (*Figure 5E*).

## TRIM5 binding to HIV-1 cores

The susceptibility of different retroviruses to restriction by different TRIM5 variants can vary dramatically and appears to be determined largely at the level of capsid recognition (*Li et al., 2006*; *Ohkura et al., 2006*; *Perez-Caballero et al., 2005*; *Sebastian and Luban, 2005*; *Song et al., 2005a*; *Stremlau et al., 2004*; *2005*; *2006*). Consistent with previous reports (*Sayah et al., 2004*; *Song et al., 2005b*; *Stremlau et al., 2004*), we found that TRIM5 proteins restricted the transduction of HeLa cells with an HIV-1 reporter vector, and the strength of restriction followed the order: TRIMCyp and TRIMCyp$_{K283D,Q287D}$>TRIM5α$_{rh}$>TRIM5α$_{AGMpyg}$, with no restriction observed for

TRIM5$\alpha_{cpz}$ or TRIM5$\alpha_{hu}$ (*Figure 2—figure supplement 1*). A sucrose cushion co-sedimentation assay was again used to test whether pure recombinant TRIM5$\alpha$ proteins bound directly to hyperstable HIV-1 cores, and whether core binding correlated with restriction activity. These experiments were performed with two proteins that restrict HIV-1, TRIM5$\alpha_{AGMpyg}$ and TRIM5$\alpha_{rh}$, and one that does not, TRIM5$\alpha_{cpz}$.

As shown in *Figure 6A*, the two restricting TRIM5$\alpha_{AGMpyg}$ and TRIM5$\alpha_{rh}$ proteins both co-pelleted with hyperstable HIV-1 cores and did not pellet in the absence of cores (compare lanes 1 vs. 2 and lanes 3 vs. 4). In contrast, the non-restricting TRIM5$\alpha_{cpz}$ protein did not bind cores under the same conditions (compare lanes 5 vs. 6). These core binding experiments were performed in the presence of excess TRIM5$\alpha$ proteins, and the approximate stoichiometry of the pelleted core-TRIM5$\alpha_{AGMpyg}$ complexes was estimated by comparing the levels of CA and TRIM5$\alpha_{AGMpyg}$ to standard curves of known protein concentrations. The measured TRIM5$\alpha_{AGMpyg}$:CA ratio in these experiments was 1:7 ± 2 (n = 4). Based upon their relative sizes, we estimate that each TRIM5 ring will cover ~14 CA hexamers (*Ganser-Pornillos et al., 2011*). On the basis of an idealized binding model and the known stoichiometries of each ring (CA = 6, TRIM5$\alpha$ = 12), we estimate that a fully saturated capsid would have a TRIM5:CA ratio of ~1:14. Hence, two different restricting TRIM5$\alpha$ proteins can bind directly to hyperstable HIV-1 cores at near saturating levels in vitro, whereas the non-restricting TRIM5$\alpha_{cpz}$ protein does not bind cores. Consistent with the solution binding experiments, negative stain electron microscopic images again revealed thin ring-like assemblies of TRIM5$\alpha_{rh}$ proteins on the capsid surfaces of hyperstable viral cores (*Figure 6B*, compare the cores in the second and third rows with the undecorated cores in the first row).

## TRIM5 decorated HIV-1 cores

Free and TRIM5$\alpha_{AGMpyg}$-decorated cores were also visualized in three dimensions by ECT (*Figure 7* and *Figure 7—figure supplement 1*). Free hyperstable cores could be spherical, cylindrical, or conical, and general size distributions and lattice features were similar to native HIV-1 cores (*Figure 7—figure supplement 1*). Holes in the tips of some conical cores were also observed, supporting previous results suggesting that cores are frequently unclosed (*Yu et al., 2013*).

Samples in which cores were incubated with TRIM5$\alpha_{AGMpyg}$ typically exhibited networks of densities at the air/water interface. These networks exhibited three-fold vertices in the plane of the interface that were similar to 2D hexagonal TRIM5$\alpha$ lattices but did not exhibit diffraction or crystalline order (*Video 2*). Approximately 12% of cores imaged were also decorated on their outer capsid surfaces with TRIM5$\alpha_{AGMpyg}$ densities. Multiple extended density 'arms' approximately 19 nm in length were seen arranged in a roughly hexagonal pattern (*Figure 7A*, and *Video 2*).

The prevalence of TRIM5$\alpha_{AGMpyg}$ at the air/water interface and relative paucity of TRIM5$\alpha$-decorated cores suggested that previously core-bound TRIM5$\alpha$ was being lost to the air/water interface during plunge-freezing (*Video 3*). To reduce this problem, we crosslinked the core-TRIM5$\alpha$ complexes with ethylene glycol bis(sulfosuccinimidylsuccinate) (Sulfo-EGS) prior to plunge freezing. Even after crosslinking, a substantial amount of TRIM5$\alpha$ was still seen at the air/water interface, but now the majority of cores were decorated with TRIM5$\alpha_{AGMpyg}$. Analysis of the volume surrounding these cores revealed broken, but extensive TRIM5$\alpha$ hexagonal nets, which in some cases enveloped the entire capsid (*Figure 7B*, *Figure 7—figure supplement 2*, and *Videos 4*). TRIM5$\alpha$ nets on bona fide cores were irregular, but exhibited a similar distribution of hexamer edge lengths (20 ± 2 nm, n = 51) and vertex angles (120 ± 2°, n = 68) as CA/TRIM5 'co-assemblies' (*Figure 7—figure supplement 3*). In some cases, TRIM5$\alpha$ densities with a length of ~38 nm were observed on the capsid surface (data not shown). These longer densities might occur through the association of two sets of TRIM5$\alpha$ dimers via dimeric rather than trimeric B-box 2 domain interactions (*Wagner et al., 2016*). Despite these irregularities, we were able to perform coarse, rigid body fitting of the crystal structures of B-box 2 trimers and coiled-coil dimers into 3D density maps. As shown in *Figure 7C*, each edge was occupied with a TRIM5$\alpha$ coiled-coil dimer and the three-fold densities were occupied by three B-box 2 domains. The fitting revealed that TRIM5 hexagonal lattices are in good agreement with the crystal structures in dimensions and angles. Overall, our results indicate that TRIM5 proteins form flexible hexagonal nets on the capsid surface with their domain positions schematically shown in *Figure 7D*.

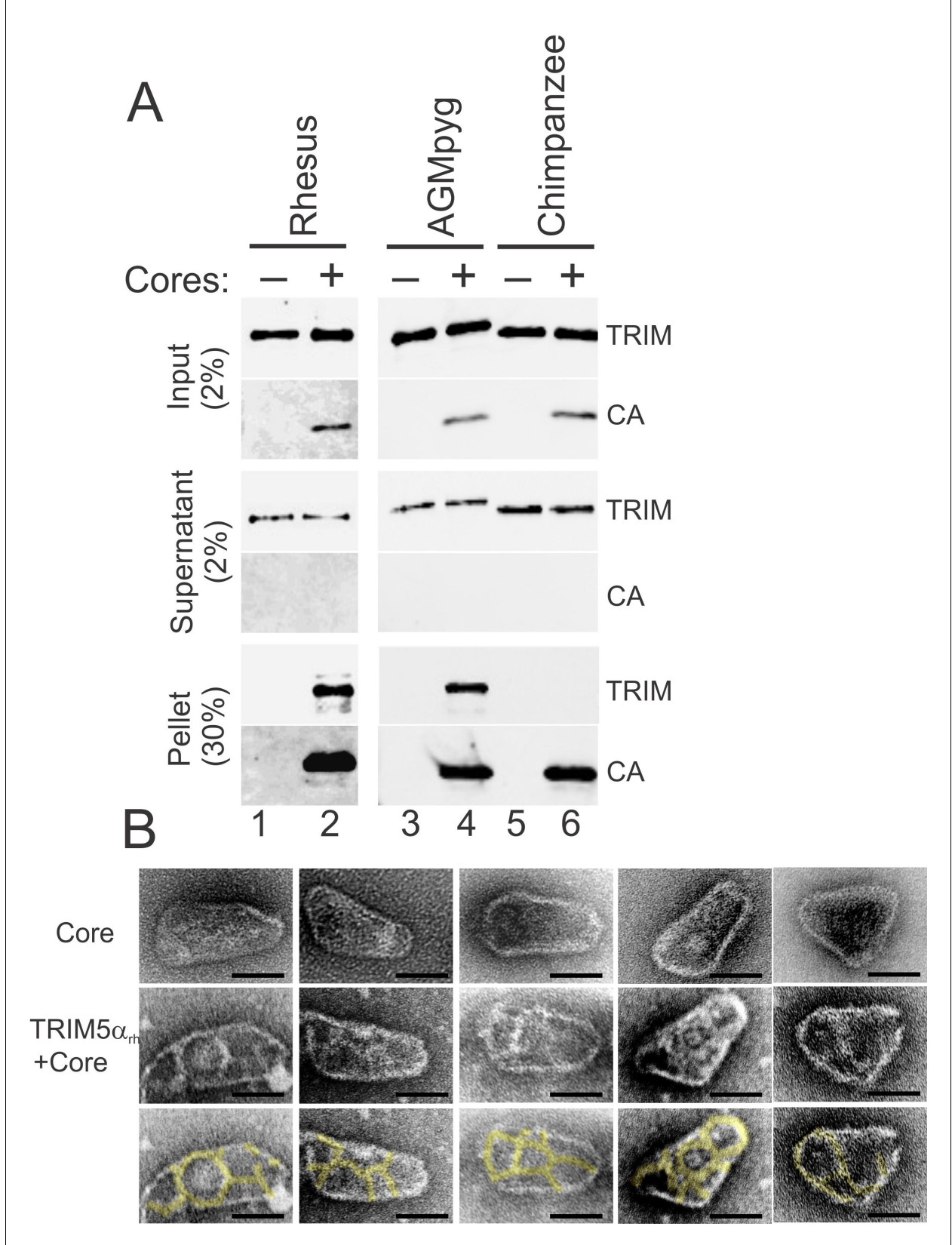

**Figure 6.** TRIM5α proteins bind directly to HIV-1 cores. (**A**) Sucrose cushion co-sedimentation binding assay for TRIM5α-HIV-1 core interactions. TRIM5α proteins were incubated in the absence of cores (lanes 1, 3, 5) or in the presence of hyperstable HIV-1 cores (lanes 2, 4, 6), and the mixtures

*Figure 6 continued on next page*

*Figure 6 continued*

were subjected to centrifugation through the sucrose cushion. Pelletable cores and bound TRIM5α (Pellet, 30% of total) and unbound TRIM5α (Supernatant, 2% of total) were analyzed by western blotting for TRIM5α and CA proteins. The input levels of both proteins are also shown for reference (Input, 2% of total). Representative results from one of three independent experiments are shown. (B) Representative electron micrographs of control HIV-1 cores (first row), and cores decorated with TRIM5α$_{rh}$ (second row) and negatively stained with uranyl acetate. TRIM5α decorations were highlighted in yellow (third row). Scale bars are 50 nm.

## Discussion

Our studies further support the prevailing models that TRIM5 restriction factors bind directly to the surfaces of incoming retroviral capsids and that restriction susceptibility is dictated at the level of capsid recognition (*Li et al., 2006*; *Ohkura et al., 2006*; *Perez-Caballero et al., 2005*; *Sebastian and Luban, 2005*; *Song et al., 2005a*; *Stremlau et al., 2004*; *2005*; *2006*). In addition, we find that the ability to assemble into hexagonal nets comprising open, six-sided rings is a conserved feature of multiple different primate TRIM5 proteins. Our EM analyses, together with recent crystal structures of fragments of non-assembling TRIM5 proteins that span the core coiled-coil and L2 linker regions (*Goldstone et al., 2014*; *Sanchez et al., 2014*), indicate that each ring edge is formed by a TRIM5 dimer that displays two SPRY (or CypA) recognition domains at its center. Most importantly, we find that capsid binding and TRIM5 assembly are coupled processes that cooperate to promote the recognition of pleomorphic retroviral cores with high affinity and specificity.

### Reagent development

Through the course of our studies we developed and characterized two new sets of reagents for studying retroviral replication and restriction: hyperstable, disulfide-crosslinked HIV-1 capsids and pure recombinant primate TRIM5 proteins. The generation of hyperstable capsids was enabled by previous studies showing that Cys residues at CA positions 14 and 45 form disulfide bonds efficiently in vitro when these residues are closely juxtaposed within the CA hexamer (*Pornillos et al., 2009*; *2010*). Our experiments demonstrate that these disulfides also form efficiently in the context of the intact HIV-1 capsid. A similar disulfide crosslinking strategy was also used to link CA trimers at cysteine positions 207 and 216 across the local three-fold axes of the HIV-1 capsid (*Byeon et al., 2009*; *Zhao et al., 2011*). We have compared these two different systems, as well as an alternative strategy in which CA hexamers were crosslinked by disulfide bonds between Cys residues at CA positions 42 and 54 (*Pornillos et al., 2010*). Disulfide bonds form readily within viral capsids in all three cases (*Byeon et al., 2009*; *Pornillos et al., 2009*; *2010*; *Zhao et al., 2011*) (and data not shown), and we anticipate that the different crosslinking strategies could have distinct advantages depending upon the application. For example, we have confirmed the report that HIV-1 cores with trimerized CA proteins retain modest infectivity (*Byeon et al., 2009*), whereas infectivity is almost completely abolished when cores are crosslinked at either site in the CA hexamer. We also find, however, that viral core yields and stabilities are greater for the two hexamer crosslinking systems, and are highest for the Cys14/Cys45 system described here. Thus, these hyperstable capsids should be optimal for analyzing the binding of a series of proteins and drugs that have recently been described, including the proteins CPSF6 (*Bhattacharya et al., 2014*; *Lee et al., 2010*; *Price et al., 2014*), Nup153 (*Bhattacharya et al., 2014*; *Di Nunzio et al., 2013*; *Matreyek et al., 2013*; *Price et al., 2014*), and Nup358 (*Bichel et al., 2013*; *Meehan et al., 2014*; *Schaller et al., 2011*), and the inhibitors PF-74 (*Bhattacharya et al., 2014*; *Blair et al., 2010*; *Fricke et al., 2013*; *Price et al., 2014*) and BI-1 and BI-2 (*Fricke et al., 2014*; *Lamorte et al., 2013*; *Price et al., 2014*). Hyperstable capsids may also represent a useful starting point for the development of in vitro viral replication assays, particularly if the capsid disulfides can be reduced without inactivating the internal reverse transcriptase and integrase enzymes.

The development of systems for producing milligram quantities of pure recombinant primate TRIM5 proteins should similarly facilitate studies of restriction by advancing methods for protein detection and enabling new mechanistic and structural analyses. For example, our recombinant TRIM5α$_{rh}$ proteins have already been used successfully as antigens to generate monoclonal antibodies that can detect endogenous TRIM5α proteins (NIH AIDS Reagent Program and (*Imam et al., 2016*). Moreover, although structural studies of TRIM5 protein domains and fragments have made

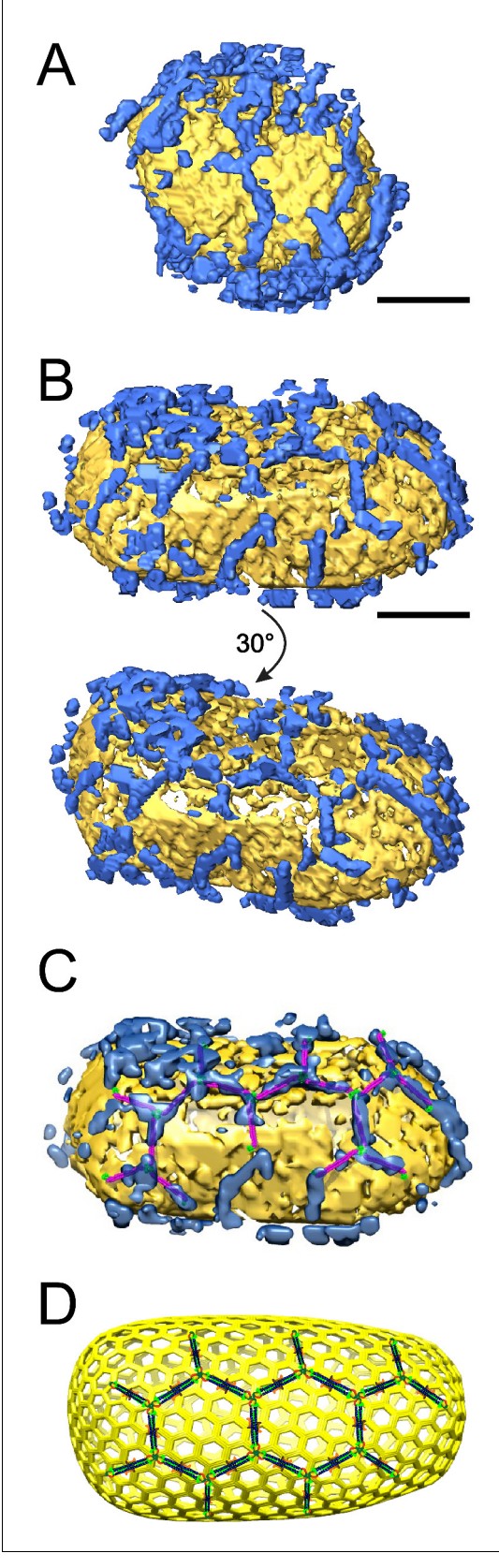

**Figure 7.** ECT reveals that TRIM5α forms flexible hexagonal nets on hyperstable HIV-1 cores. (**A**) *Figure 7 continued on next page*

valuable contributions to our understanding of TRIM5 structure and enzymology, there are a number of indications that the different domains work together as an integrated machine (*Ganser-Pornillos et al., 2011*; *Goldstone et al., 2014*; *Li et al., 2013*; *Li and Sodroski, 2008*; *Reymond et al., 2001*; *Sanchez et al., 2014*). It will therefore also be important to study intact TRIM5 proteins, particularly to determine how capsid recognition is coupled to ubiquitin signaling (*Fletcher et al., 2015*; *Pertel et al., 2011*; *Yudina et al., 2015*).

## TRIM5 recognition of retroviral capsids and its implications for restriction

Antiviral innate immune factors that function by recognizing retroviral capsids must overcome considerable sequence and structural variability. Primate TRIM5 proteins accomplish this task by coupling weak recognition of conserved capsid epitopes with hexagonal net assembly, thereby amplifying intrinsically weak binding affinities through avidity effects (*Ganser-Pornillos et al., 2011*; *Price et al., 2009*). Our studies confirm that hexagonal net assembly is a conserved property, but also reveal that the TRIM5 hexagonal nets are not highly regular. The lack of strict regularity in the TRIM5 net may be required to adapt to the lack of regularity in the opposing retroviral capsids, where every CA hexagon occupies a slightly different local environment and where pentagons and other kinds of lattice 'defects' are also prevalent (*Byeon et al., 2009*; *Ganser-Pornillos et al., 2011*; *Gres et al., 2015*; *Hatziioannou et al., 2004*; *Obal et al., 2015*; *Pornillos et al., 2011*; *Yu et al., 2013*; *Zhao et al., 2013*).

Consistent with a lattice assembly model, imaging studies have provided direct evidence that multiple TRIM5 molecules can bind continuously to incoming capsids (*Campbell et al., 2008*; *Danielson et al., 2012*; *Lukic et al., 2011*). The stoichiometry of TRIM5-capsid interactions within cells is not yet known, but we find that TRIM5 molecules can cover most of the capsid surface in vitro. A patch of just 4–6 TRIM5 rings covers about half of the capsid surface (see *Figure 7*), however, and would present ~40 recognition domains for avid capsid binding. Thus, the entire capsid probably does not need to be completely enveloped within a surrounding TRIM5 lattice for restriction to occur. Indeed, CA mixing studies have

*Figure 7 continued*

Segmented HIV-1 core (yellow) decorated with TRIM5α (blue). (B) Segmented HIV-1 core decorated with TRIM5α and subjected to mild sulfo-EGS crosslinking prior to vitrification. (C) TRIM5α structural model docked into the cryoEM volume of the tomogram shown in (B). (D) Idealized schematic model of an HIV-1 fullerene cone bound by a TRIM5α hexagonal net. Domains and linkers of TRIM5α are colored as described in *Figure 1A*. Scale bars are 35 nm.

The following figure supplements are available for figure 7:

**Figure supplement 1.** ECT of hyperstable HIV-1 cores.

**Figure supplement 2.** ECT of a HIV-1 hyperstable core in complex with TRIM5α.

**Figure supplement 3.** Characterization of TRIM5 hexagonal nets on HIV-1 cores.

shown that efficient TRIM5 restriction can occur even when only 25% of the capsid subunits are competent for TRIM5 binding (*Shi et al., 2013*).

TRIM5-mediated restriction appears to proceed via a multi-step pathway in which capsid recognition is followed by steps that lead to capsid dissociation and inhibition of reverse transcription (*Pertel et al., 2011*; *Stremlau et al., 2006*). In cells, the later processes can be decoupled from the initial binding event by treatment with proteasome inhibitors or by mutations in the RING domain (*Anderson et al., 2006*; *Fletcher et al., 2015*; *Kutluay et al., 2013*; *Roa et al., 2012*; *Wu et al., 2006*). These treatments do not block TRIM5 binding, but presumably do interfere with ubiquitin-mediated signaling events. Thus, although TRIM5 binding can destabilize helical CA tubes in vitro (*Black and Aiken, 2010*; *Zhao et al., 2011*), capsid dissociation and inhibition of reverse transcription appear to require ubiquitin-dependent signaling in

cells (*Campbell et al., 2015*; *Fletcher et al., 2015*). Capsid binding can activate TRIM5 ubiquitin E3 ligase activity in vitro (*Pertel et al., 2011*), and the structure of the hexagonal TRIM5 lattice suggests how this could occur. Recent structural studies of the RING domains from TRIM37 (PDB ID: 3LRQ) and TRIM5α (PDB ID: 4TKP) (*Yudina et al., 2015*) have demonstrated that the RING domains function as dimers. However, the antiparallel structure of the TRIM5 coiled-coil precludes close contact of the two RING domains within a single TRIM5 dimer (*Goldstone et al., 2014*; *Sanchez et al., 2014*). Thus, RING domains from multiple different TRIM5 dimers apparently must come together to transfer ubiquitin. Our lattice structures reveal such associations of three TRIM5 RING domains at local three-fold axes in the hexagonal net (*Figure 7D*). This suggests that two of the RING domains may join to form an active dimer and that the third 'orphan' RING domain could then be used as a substrate for autoubiquitylation. This idea is supported by 1) recent studies that demonstrate that autoubiquitylation occurs in vitro and in cells and that Lys45 and Lys50 within the RING domain of rhesus TRIM5α are preferentially ubiquitylated (*Fletcher et al., 2015*), and 2) the accompanying structural studies (*Wagner et al., 2016*) of a truncated rhesus TRIM5α protein comprising a B-box 2 and the truncated coiled-coil and L2 linker domains (termed 'mini-TRIM'), which reveal that the B-box 2 domains can

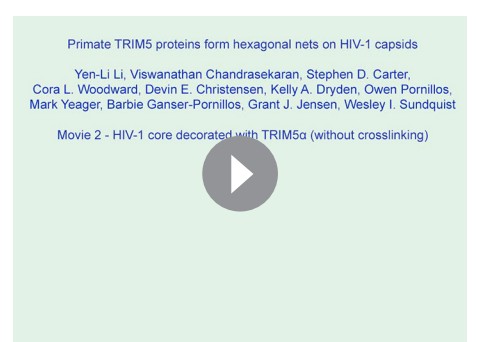

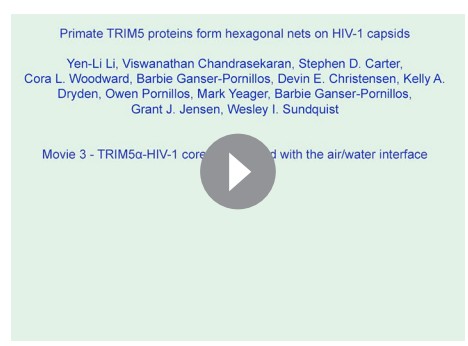

**Video 2.** HIV-1 core decorated with TRIM5α (without crosslinking). ECT of HIV-1 core (yellow) decorated with TRIM5α (blue). Video corresponds to *Figure 7A*.

**Video 3.** TRIM5α-HIV-1 cores associated with the air/water interface. Representative ECT showing the interaction of TRIM5α-HIV-1 cores with the air/water interface. Video corresponds to *Figure 7*.

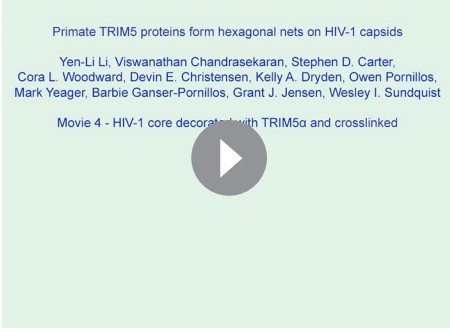

**Video 4.** HIV-1 core decorated with TRIM5α and crosslinked. HIV-1 core (yellow) decorated with TRIM5α (blue) were subjected to crosslinking and imaged by ECT. Video corresponds to *Figure 7B and C*.

mediate trimerization. Hence, in addition to enhancing binding avidity, formation of hexagonal TRIM5 lattices may also activate the ubiquitin signaling cascade that ultimately results in capsid dissociation and inhibition of reverse transcription.

## Materials and methods

### Plasmids, cells and antibodies

HEK 293T and HeLa cells were grown at 37°C with 5% $CO_2$ in DMEM media (Gibco) supplemented with 10% heat-inactivated fetal calf serum and 2 mM L-glutamine. Plasmid constructs for virus production and for expressing TRIM5, CA, and OSF-CypA in mammalian, insect and bacterial cells were created by standard cloning and mutagenesis methods (details available upon request). The plasmids used in this study are summarized in *Supplementary file 1A*. All plasmids have been submitted to the Addgene (https://www.addgene.org/) and DNASU (https://dnasu.org/DNASU/) public repository.

### TRIM5-21R electron cryo-tomography (ECT)

TRIM5-21R and TRIM5-21R$_{\Delta SPRY}$ proteins were expressed, purified and assembled into 2D crystals as previously described (*Ganser-Pornillos et al., 2011*), except that the TRIM5-21R$_{1-300}$ assembly was promoted by addition of an equal volume of 0.1 M sodium chloride, 0.1 M bicine, pH 9.0, 20% polyethylene glycol monomethyl ether 5000 to the concentrated protein solution.

To prepare samples for ECT, 3 µl of polymerized lattice was mixed with 10 nm Au fiducials and applied to a 2/2 holey carbon-coated Cu EM grid (Quantifoil) and transferred with forceps to the environment chamber of a Vitrobot Mark III (FEI) maintained at 25°C and 80% relative humidity. Excess liquid was manually blotted from the grids on one side before plunging into liquid ethane. Cryo-preserved grids were imaged in a 300 kV FEI G2 Polara equipped with a field emission gun and energy filter (slit width set at 20 eV), and fitted with a K2 Summit direct detector. Tilt-series were collected over a series of angles ranging from −60° to +60° using a step size of 1°; 22,500x magnification (effective pixel size of raw data is 5 Å), a total dose of 150 e/Å$^2$, and a defocus of -6 µm. UCSF Tomo (*Zheng et al., 2007*) was used to collect the tilt series, and 3D reconstructions were carried out using a weighted back-projection algorithm tracking 10 nm fiducials in IMOD (*Kremer et al., 1996*). The pixel size in the final reconstruction was 20 Å.

Subtomogram averages of the TRIM5-21R and TRIM5-21R$_{\Delta SPRY}$ lattices were generated using PEET in IMOD (*Nicastro et al., 2006*). 153 and 75 vertices were selected in the TRIM5-21R and TRIM5-21R$_{\Delta SPRY}$ 2D lattices, respectively, and a volume of 60 nm x 60 nm x 20 nm (x,y,z) centered on the refined positions of the selected vertices was used to generate the averaged volume. To localize the SPRY domain in the full-length TRIM5-21R lattice, the density values of the averaged lattice volumes were rescaled to reflect a mean value of zero and standard deviation of 10. The volumes were then aligned in Chimera (*Pettersen et al., 2004*), and the TRIM5-21R$_{\Delta SPRY}$ density values were subtracted from the TRIM5-21R volume. The resulting density difference map was contoured and displayed at 3 sigma above the mean.

An important difference between the TRIM5-21R and TRIM5-21R$_{\Delta SPRY}$ assemblies is that in the absence of the SPRY domains, multiple hexagonal lattices stacked on 'top' of one another close together, laterally offset by a quarter, half, or three-quarters the distance across a hexagon. Thus in projections through 3D subtomogram averages, like those shown in the bottom middle panels of *Figure 1C and D*, all of these other offset lattices appear, but less prominently than the main lattice. Due to the special pattern of the offsets (one quarter, half, and three-quarters across) and the hexagonal geometry, their projections all intersect at the center of the arms of the main lattice, causing that position to appear especially dark and large. Furthermore, the four-helix bundle of both TRIM5-

21R and TRIM5-21R$_{\Delta SPRY}$ also contributed to the observed densities in the center of the hexagon edges. *Figure 1E* shows, however, a thin slice through the 3D difference map containing only the main hexagonal lattice, revealing the position of the SPRY domain without interference from the lattices above and below.

To quantify the variability in the TRIM5-21R hexagonal lattice, the refined positions of each vertex were used to calculate: 1) the distance between neighboring vertices, and 2) the average angle of hexamer edges extending from the three-fold vertices. These values were entered into the *imodsetvalues* program in IMOD and a pseudo-colored model was generated to reflect length (colored lines) and average angles (colored spheres) (*Figure 1F*).

## TRIM5 restriction assays

HEK 293T cells were used to generate lentiviral vectors for transduction of HeLa cells for expression of TRIM5 proteins with a C-terminal Flag One-STrEP tag. pCMV-ΔR8.2 (structural genes) (*Naldini et al., 1996*), pCMV-VSVG (envelope) (*Sandrin et al., 2002*; *Yee et al., 1994*), and CSII-IDR2 (contains a packaging signal and genes for TRIM5 and DsRed) were co-transfected in 293T cells. After 3 days, virion-containing media was removed from the cells, passed through a 0.45 μm filter (Nalgene SFCA syringe filters), layered on top of a 20% sucrose cushion in HS buffer (10 mM HEPES pH 7.2 and 140 mM NaCl) and spun in an Optima L-90K Ultracentrifuge at 96,281 g (Beckman SW32 Ti rotor) for 2 hr at 4°C. Virion-containing pellets were resuspended in HS buffer, aliquoted, and frozen at -80°C. Thawed aliquots were titrated on HeLa cells to determine viral titers by monitoring the number of DsRed positive cells using fluorescence-activated cell sorting (FACS).

HeLa cells (1 x 10$^5$ cells per well of 6-well plate) were transduced with lentiviral vectors expressing different TRIM5 proteins at an multiplicity of infection (MOI) of 1. Three days after transduction, cells were split and reseeded at 5 x 10$^4$ cells per well of a 24-well plate and infected with increasing amounts of HIV-GFP per well. The remaining cells were used for western blot analysis to determine TRIM5 expression levels. Three days after infection with HIV-GFP, cells were trypsinized, and GFP and DsRed positive cells were counted using FACS. Only DsRed positive cells (which also expressed TRIM5) were used for statistical analysis of HIV-GFP restriction.

## Expression and purification of native TRIM5 proteins

Recombinant baculoviruses expressing TRIM5 proteins with either N-terminal One-STrEP-FLAG (OSF) or C-terminal FLAG-One-STrEP (FOS) HRV14-3C protease-cleavable tags were generated using the Bac-to-Bac baculovirus expression system (Thermo Fisher Scientific). Suspension SF9 insect cells (2 L at 2 x 10$^6$ cells/ml) grown in ESF-921 medium (Expression Systems) were infected with recombinant baculoviruses at an MOI of 10, and harvested by centrifugation 48 hr later. All purification steps were performed at 4°C. Cell pellets were resuspended in 5 times the pellet volume of lysis buffer (70 mM N-Cyclohexyl-2-aminoethanesulfonic acid (CHES), 100 mM NDSB-256, 1.5% Triton X-100, 100 nM ZnCl$_2$, 1 mM Tris(2-carboxyethyl)phosphine (TCEP), 0.7% protease inhibitor cocktail (v/v, Sigma), 100 U avidin, pH 10.0) and lysed by freeze-thaw and sonication (3 x 30 s on ice; Branson sonifier 450, 50% duty cycle, 50% output). Cell lysates were clarified by ultracentrifugation at 184,000 g (Beckman Ti 50.2 rotor) for 1 hr. The supernatants were filtered (0.45 μm) and loaded onto a 5 ml StrepTrap HP column (GE Healthcare) pre-equilibrated in binding buffer (20 mM CHES, 100 nM ZnCl$_2$, 1 mM TCEP, pH 10.0). The column was washed with 20 column volumes (CV) of binding buffer supplemented with 1 M NaCl and 100 U avidin (VWR), followed by 5 CV of binding buffer. The protein was eluted in 6 CV binding buffer supplemented with 2.5 mM D-desthiobiotin (Sigma). The eluate was diluted to 0.3 mg/ml protein in binding buffer to minimize protein loss due to self-assembly, and dialyzed overnight against 1 L cleavage buffer (25 mM Tris, 1 mM TCEP, pH 8.0) supplemented with ~ 1:100 (by mass, enzyme:substrate) His$_6$-HRV14-3C and His$_6$-Usp2 enzymes to remove the OSF tag and any linked ubiquitin added during insect cell expression. TRIM5α$_{hu}$ and TRIMCyp formed soluble/insoluble aggregates at pH 8.0 and were therefore dialyzed against 20 mM CHES, 1 mM TCEP, pH 9.0. Most TRIM5 proteins were sensitive to non-specific internal proteolysis by HRV14-3C protease. We therefore used the minimal amount (which differed between constructs) required to completely cleave the OSF tag overnight. When cleavage was complete, the pH of the protein solution was adjusted to 10 by direct addition of 1 M CHES, pH 10.0, to a final concentration of 100 mM. The sample was applied onto two tandem 5 ml HiTrap Q HP columns (GE

Healthcare) pre-equilibrated with binding buffer, and eluted with a 12 CV linear NaCl gradient (0–1 M) in binding buffer. Fractions containing TRIM5 proteins were pooled, dialyzed against 1 L binding buffer for at least 4 hr, loaded onto a HiLoad 16/600 Superdex 200 gel filtration column (GE Healthcare) pre-equilibrated with binding buffer, and eluted in 1 CV of binding buffer. Fractions corresponding to TRIM5 dimers were pooled and concentrated to 1 mg/ml using a Vivaspin 20 concentrator (10,000 MWCO PES for TRIM5$\alpha_{AGMpyg\Delta SPRY}$ and 30,000 MWCO PES for full-length TRIM5$\alpha$ and TRIMCyp, Sartorius Stedim). Average yields were 4 mg (1.3–9.6 mg) per liter insect cell culture and protein identities were confirmed by electrospray ionization mass spectrometry (ESI-MS) (see *Supplementary file 1B*).

## TRIM5 2D crystallization

Freshly purified TRIM5$\alpha_{AGMpyg}$ protein was concentrated to 1–3 mg/ml and assembled by incubating at 23°C for 1 hr and then at 4°C for 1–2 days. For EM analyses, 5 µl sample aliquots were incubated on carbon-coated EM grids for 5 min. The grids were washed by placing each grid on a single 40 µL drop of 0.1 M KCl for 3 min, briefly blotted, and then stained on a single 20 µL drop of 2% uranyl acetate for an additional 3 min.

Unlike TRIM5$\alpha_{AGMpyg}$, TRIMCyp did not spontaneously assemble into hexagonal lattices following concentration. However, crystals were occasionally observed when an equal volume of 0.01 M cobalt chloride hexahydrate, 0.1 M MES monohydrate pH 6.5, and 1.8 M ammonium sulfate was added to freshly concentrated protein at ~1 mg/ml.

## Templated assembly of TRIM5 on hexagonal arrays of HIV-1 CA-NC

As previously described (*Ganser-Pornillos et al., 2011*), 2-dimensional crystals composed of cross-linked, hexagonal HIV-1 CA were prepared by incubating 232 µM CA-NC$_{A14C/E45C/W184A}$ with a small 25-TG oligo (143 µM). TRIM5 proteins were then added in 1- to 10- fold molar excess, and the pH was immediately adjusted to 9.0 by direct addition of Tris buffer to a final concentration of 100 mM. Samples were incubated for 1–96 hr, applied to carbon-coated EM grids for 60 s, washed and stained as described above, and visualized by EM. Ten fold lower amounts were sufficient for TRIM-Cyp$_{K283D,Q287D}$ templated assembly.

## Bacterial protein expression and purification

### HIV-1 CA$_{A14C/E45C/A92E}$

2L of *E.coli* Rosetta (λDE3) pLysS cells (Stratagene) carrying the HIV-1 CA A14C/E45C/A92E expression construct were grown to an OD$_{600\ nm}$ of 0.6 in LB medium at 37°C, cooled to 19°C and protein expression was induced with 1 mM isopropyl-β-D-thiogalactopyranoside (IPTG) followed by overnight incubation with shaking. HIV-1 CA proteins were purified and assembled into tubes as previously described (*Pornillos et al., 2010*), except that a higher concentration of dithiothreitol (DTT, 100 mM) was used during protein purification to improve solubility and increase yields. All purification steps were performed at 4°C. Cells were lysed as described above, and proteins were purified from clarified lysates by ammonium sulfate precipitation, dialyzed against 25 mM MOPS, 100 mM DTT, pH 6.5, and loaded onto a 5 ml HiTrap Q HP column (GE Healthcare) pre-equilibrated with the same buffer. The flow-through was applied onto a 5 ml Hi-Load SP Sepharose High Performance column (GE Lifesciences) and eluted with a linear NaCl gradient (0–500 mM) in the same buffer. Fractions containing CA proteins were pooled and dialyzed overnight against storage buffer (20 mM Tris, 40 mM NaCl and 100 mM DTT, pH 8.0). The CA proteins were then concentrated to a stock of 3 mg/ml using a Vivaspin 20 concentrator (10,000 MWCO PES, Sartorius Stedim) and stored at -80°C. Yields were ~20 mg per liter of culture, and the protein identity was confirmed by ESI-MS (MW$_{exp}$=25,667 Da, MW$_{calc}$=25,667 Da).

### OSF-cyclophilin A (OSF-CypA)

OSF-CypA was expressed in 2L of *E. coli* Rosetta (λDE3) pLysS cells (Stratagene) grown in ZYP-5052 media using an autoinduction system (*Studier, 2005*). Cells were lysed on ice by sonication in lysis buffer (50 mM Tris, pH 8.0, 50 mM NaCl, 10 mM β-mercaptoethanol (β-ME), 0.2% (w/v) deoxycholate, 2.5 nmol avidin, 20 µg/ml DNase 1) supplemented with protease inhibitors (20 µg/ml PMSF, 0.4 µg/ml pepstatin, 0.8 µg/ml leupeptin and 1.6 µg/ml aprotinin). All purification steps were

performed at 4°C. Cell lysates were clarified by centrifugation at 17,649 g (Beckman JA-20 rotor) for 45 min, filtered (0.45 µm) and loaded onto two 5 ml tandem StrepTrap HP columns (GE Healthcare) pre-equilibrated in binding buffer (100 mM Tris, 150 mM NaCl, 10 mM β-ME, pH 8.0). The column was washed with 10 CV of binding buffer, and OSF-CypA was eluted in 3 CV of the same buffer supplemented with 2.5 mM D-desthiobiotin. The protein was dialyzed against Q buffer (50 mM Tris, 50 mM NaCl, 10 mM β-ME, pH 8.0) and loaded onto two tandem 5 ml HiTrap HP Q-Sepharose anion exchange columns (GE Healthcare) pre-equilbrated in the same buffer. The OSF-CypA-containing flowthrough was collected and concentrated using Amicon Stirred Ultrafiltration Cells (Millipore). OSF-CypA (>99% pure) was obtained in high yields (~100 mg per liter bacterial culture), and its identity was confirmed by ESI-MS ($MW_{exp}$ =23,806 Da, $MW_{calc, -Met1}$=23,807 Da).

## Assembly of hyperstable $CA_{A14C/E45C/A92E}$ tubes

$CA_{A14C/E45C/A92E}$ tubes were assembled at 1 mg/ml by dialysis against dialysis buffer (20 mM Tris, pH 8.0, 1 M NaCl, and 100 mM DTT) at 4°C overnight, followed by dialysis against the same buffer lacking DTT overnight to allow the formation of disulfide crosslinks within the CA hexamers. Disulfide-crosslinked CA tubes were then dialyzed against 20 mM Tris, 40 mM NaCl, pH 8.0 and stored at 4°C.

## TRIM5-CA tube binding experiments

TRIM5-CA tube binding experiments were performed as previously described, with minor modifications (*Ganser-Pornillos et al., 2011*; *Langelier et al., 2008*; *Stremlau et al., 2006*). Recombinant TRIM5α and TRIMCyp proteins (0.25 µM) were incubated alone or with $CA_{A14C/E45C/A92E}$ tubes (2 µM) in binding buffer (20 mM HEPES, 25 mM NaCl, 1 mM TCEP, pH 7.2) in a final volume of 225 µl at 4°C for 1 hr. Aliquots (10 µl) of the incubation mixtures were mixed with 2X SDS-PAGE sample loading buffer for assessment of protein amounts in the inputs. Aliquots (200 µl) of the mixtures were layered onto a 60% (w/v) sucrose/PBS cushion (4 ml, prepared in binding buffer lacking TCEP) and subjected to centrifugation at 108,109 g (Beckman SW50.1 rotor) for 30 min at 4°C to separate free TRIM5α or TRIMCyp and unassembled CA proteins from CA tube-bound TRIM5 proteins and pelletable CA tubes. Following centrifugation, aliquots (45 µl) of supernatant (500 µl in total) were mixed with 4X SDS-PAGE sample loading buffer, and the pellets were resuspended in 25 µl 1X SDS-PAGE sample loading buffer. The TRIM5 and CA proteins in the input (3%), supernatant (3%) and pellet (30%) were separated by 12% SDS-PAGE, electrophoretically transferred onto nitrocelluose membranes (Bio-Rad) and analyzed by western blotting with mouse anti-TRIM5α monoclonal (clone 5D5-1-1), NIH AIDS Research and Reference Reagent Program, 1:000 dilution) and rabbit anti-HIV-1 CA polyclonal (made in-house UT 416, 1:3000 dilution) antibodies. Secondary IRDye800cw-conjugated donkey anti-mouse IgG (1:10,000, Rockland) or IRDye700DX-conjugated donkey anti-rabbit IgG (1:10,000, Rockland) antibodies were visualized using an Odyssey infrared imaging system (LI-COR Bioscience).

## Negative stain transmission electron microscopy

3.5 µl sample solutions of undecorated or TRIM5-decorated CA tubes were spread onto the carbon side of freshly glow-discharged, Formvar/Carbon-coated, 200-mesh copper grids (Electron Microscopy Sciences). The samples were incubated for 4 min, rinsed briefly by flotation on a drop of 100 mM KCl, blotted dry, stained for 2 min in filtered, saturated uranyl acetate (or 1 min in 1% phosphotungstate), blotted dry, and allowed to air dry. Samples were viewed on a JEOL JEM-1400 Plus transmission electron microscope operated at 120 kV accelerating voltage, and images were acquired as Gatan Digital Micrograph 3 (DM3) files with a Gatan Ultrascan CCD camera or on a Hitachi 7100 TEM at 75 kV accelerating voltage with a Gatan ORIUS CCD camera, and converted into JPEG images using ImageJ software (NIH Bethesda, MD, USA).

## Screening for TRIM5 decoration of CA tubes

TRIM5-CA tube complexes were prepared by incubating TRIM5α or $TRIMCyp_{K283D,Q287D}$ proteins with hyperstable CA tubes in 50 mM Tris, 8 mM NaCl buffer at 4°C. Decoration conditions were surveyed at a constant CA concentration (7.5 µM) over a range of TRIM5 concentrations (0.5–22.5 µM, corresponding to molar ratios of TRIM5 to CA of 1:16, 1:8, 1:6, 1:3, 1:1 and 3:1), pH values (8.0 and

9.0) and incubation times (4–92 hr). Conditions that produced the best TRIM5 decoration and minimal CA-free TRIM5 self-assemblies were determined by negative stain TEM imaging on a JEOL JEM-1400 Plus transmission electron microscope as described above. Image contrast was uniformly adjusted to enhance the decoration patterns of TRIM5 proteins on CA tubes using Adobe Photoshop CS5. The spacings of hexagonal TRIM5 rings were measured using ImageJ. Human TRIM5α and TRIM5α$_{R332P}$ tended to aggregate in all incubation conditions tested. These aggregates sometimes associated with CA tubes, but could be readily distinguished from ring-like decorated tubes. For scoring, images were judged blind by two independent colleagues.

## Deep-etch electron microscopy

TRIM5α-CA tube complexes were prepared by incubating 1 μM TRIM5α$_{AGMpyg}$ proteins with 8 μM hyperstable CA tubes in 50 mM Tris, pH 8.0, and 8.2 mM NaCl buffer at 4°C for 32 hr. Quick-freeze deep-etch EM was performed according to published protocols (*Heuser, 1980*). Briefly, a 3 μl droplet of TRIM5α$_{AGMpyg}$ decorated tubes was placed onto an acid cleaned, air dried 3x3 mm coverglass and covered by a 0.05 mm thick, 3 mm diameter wafer of sapphire on top. The sample was then mounted onto the freezing stage of a Heuser designed 'Slam Freezer' and frozen by forceful impact against a pure copper block, cooled to 4°K with liquid helium. Frozen samples were transferred to a liquid-nitrogen-cooled Balzers 400 vacuum evaporator. Freeze fracture occurred by popping off the sapphire top at -104°C under vacuum. Samples were etched for 210 s at −104°C and rotary replicated with ~3 nm platinum deposited from a 15° angle above the horizontal, followed by an immediate ~10 nm stabilization film of pure carbon deposited from an 85° angle. Replicas were floated onto a dish of concentrated hydrofluoric acid and transferred through 3 rinses of distilled H$_2$O containing a loopful of Photo-flo. Replicas were picked up on formvar coated copper grids, and imaged on a JEOL 1400 microscope with attached AMT digital camera.

## Co-assembly of TRIM5α and HIV-1 CA

Coassembly experiments were performed by incubating HIV-1 CA (650 μM) alone or with TRIM5α$_{AGMpyg}$ (1–15 μM) in assembly buffer (20 mM Tris, pH 8.0, 50 mM NaCl) at 37°C for 2–3 hr, followed by a 2 hr incubation at room temperature. Following incubation, a 3.5 μL aliquot of the assembly reaction was incubated on carbon-coated EM grids (Electron Microscopy Sciences) for 1–2 min. Grids were then placed directly onto a 20 μL drop of 0.1 M KCl for 2 min, blotted and moved to a 20 μL drop of 2% uranyl acetate for 2 min, blotted, and air dried. Samples were imaged on either a Tecnai T12 or a Tecnai F20 microscope operating at 120 kV.

## Preparation of HIV-1 virions

HEK 293T cells (29 x 10 cm plates) were co-transfected (polyethylenimine, PEI, Polysciences) at 70–80% confluency with pLOX-GFP (5 μg DNA/plate) (*Salmon et al., 2000*) and pCMV-ΔR8.2 vectors (5 μg DNA/plate) (*Naldini et al., 1996*) that expressed HIV structural proteins encoding wild type or mutant CA sequences (A14C/E45C or A14C/E45C/A92E). 40 hr later, virion-containing media was pooled, filtered (0.45 μm) and pelleted by ultracentrifugation through a 4 ml, 20% sucrose/PBS cushion in 25 x 89 mm polyallomer centrifuge tubes (Beckman Coulter) at 96,281 g (Beckman SW32 Ti rotor) for 2 hr at 4°C. Subsequent core purification steps were performed at 4°C.

## Sucrose gradient purification of HIV-1 cores

Wild type and hyperstable HIV-1 A14C/E45C cores were isolated from virions using an adaptation of a sucrose-gradient, spin-through method (*Kotov et al., 1999*; *Langelier et al., 2008*). Virion pellets were resuspended with 2.4 ml ST buffer (20 mM Tris, 75 mM NaCl, pH 7.4). 6 x 11.5 ml 30–70% (w/v) continuous sucrose gradients in ST buffer were made in 14 x 89 mm polyallomer centrifuge tubes (Beckman Coulter) using a gradient maker (Biocomp). The gradients were overlaid with a 300 μl 15% (w/v) sucrose cushion in ST buffer containing 0.5% Triton X-100 (to delipidate the virions as they migrated through the cushion) and then with a 300 μl non-detergent barrier layer (7.5% sucrose in ST buffer), which protected virions from premature detergent exposure. Concentrated virions were applied to the top of the gradient and subjected to centrifugation at 151,263 g (Beckman SW41 Ti rotor) for 16 hr. Twelve 1 ml fractions were collected from the bottom of each tube, and the density of each fraction was determined from the refractive index using a digital refractometer (Leica). The

CA content in each fraction was analyzed by western blotting using rabbit anti-HIV-1 CA polyclonal antibodies (made in-house, UT 416, 1:3000 dilution). Fractions 10–12 (density = 1.22–1.27 g/ml), which contained intact HIV-1 cores, were pooled, diluted with ST buffer, and subjected to ultracentrifugation at 151,263 g (Beckman SW41 Ti rotor) for 2 hr. The pelleted cores were resuspended in 240 µl ST buffer, and recovered yields were quantified as described below.

## Affinity purification of HIV-1 cores

2.4 ml of concentrated virions in PBS were mixed gently with an equal volume of lysis buffer (1% Triton X-100, 100 mM Tris, 2M NaCl, pH 8.0) in the presence of 35 µM OSF-CypA and incubated for 3 min at 23°C. Subsequent core purification steps were performed at 4°C. 8 mg of MagStrep'type2HC' beads (IBA GmbH) were added to the lysed virions and mixed gently by inversion for 7 min to allow OSF-CypA to bind the membrane-stripped cores. The sample was then placed on a PolyATtract system 1000 magnet separation stand (Promega) for 3 min, and the supernatant ('Flow-through' in *Figure 5—figure supplement 1E*) was removed. Captured cores were washed 10 times with high salt buffer (50 mM Tris, 1 M NaCl, pH 8.0) to remove unbound CA proteins and contaminating vesicles, and the final wash sample was saved for western blot analysis ('Wash' in *Figure 5—figure supplement 1E*). Cores were eluted in 150 µl elution buffer (50 mM Tris, 75 mM NaCl, pH 8.0) supplemented with 40 µM Cyclosporine A (Sigma-Aldrich) and incubated with inversion for 40 min. The sample was subjected to brief centrifugation in a tabletop ultracentrifuge at 1000 g for 5 s and placed on a Magnesphere Technology Magnetic Separation Stand (Promega) for 5 min. The supernatant containing the purified cores ('Eluate' in *Figure 5—figure supplement 1E*) was collected and used in the experiments shown in *Figures 5*, *6*, *7*, *Figure 5—figure supplement 1*, and *Figure 7—figure supplement 1*, *2* and *3*. Beads before and after CsA elution were also saved for western blot analyses (*Figure 5—figure supplement 1E*).

## Characterization of purified hyperstable cores

### Core yields

Virion inputs and core yields were quantified by western blot densitometry against a standard curve of recombinant CA proteins for reference. The recovery of cores from virions was calculated by normalizing core yields of CA to corresponding virion CA inputs, which were set to 100%. As illustrated in *Figure 5*, disulfide crosslinks apparently stabilized the HIV-1 cores, resulting in a ~4 fold increase in the core recovery (0.8 ± 1%; core yields: 0.6 ± 0.5 µg CA; virus input: 100 ± 80 µg CA, n = 10) compared to wild type cores (0.2 ± 0.1%; core yields: 0.08 ± 0.05 µg CA; virus input: 40 ± 30 µg CA, n = 7). The affinity purification method consistently raised the yield of hyperstable HIV-1 A14C/E45C cores by an additional ~4 fold (3 ± 2%; core yields: 1 ± 0.6 µg CA; virus input: 40 ± 8 µg CA, n = 3) vs. the sucrose-gradient, spin-through method. Core recovery was not affected by the A92E mutation (3 ± 1%; core yields: 3 ± 1 µg CA; virus input: 100 ± 60 µg CA, n = 7).

### Analyses of HIV-1 core morphologies

Discrete particles were imaged by negative stain EM and scored as 'tubular' if their edges appeared parallel, as 'spherical' if they were spherical or elliptical, and as 'conical' if they lacked the above properties. The final class included conical, triangular, bullet-shaped, and coffin-shaped cores.

### Disulfide crosslinks

To examine disulfide crosslinking within purified cores, sucrose gradient fractions 7–9 and 10–12 were pooled separately, mixed with SDS-PAGE sample loading buffer lacking β-ME (or containing the concentrations designated in *Figure 5C*), treated with 31.25 µM methyl methanethiosulfonate (Pierce Biotechnology), heated at 95°C for 10 min, and electrophoresis was performed on 4–15% gradient SDS polyacrylamide gels (Bio-Rad) and analyzed by western blotting.

### Analyses of HIV-1 core protein components

Purified cores were denatured in SDS-PAGE sample loading buffer, resolved on 4–15% gradient SDS polyacrylamide gels (Bio-Rad) and stained using SilverQuest silver staining kit (Thermo Fisher Scientific).

## TRIM5-core binding experiments

Recombinant TRIM5$\alpha_{AGMpyg}$ and TRIM5$\alpha_{cpz}$ proteins (0.5 µM) were incubated at 4°C for 1 hr alone or with hyperstable HIV-1 cores (0.5–1 µM) in binding buffer (40 mM HEPES, 50 mM NaCl, 1 mM TCEP, pH 7.2) in a final volume of 75 µl. TRIM5$\alpha_{rh}$ (0.25 µM) was incubated alone or with hyperstable cores under slightly more alkaline conditions (40 mM Tris, 50 mM NaCl, 1 mM TCEP, pH 8.0) to minimize untemplated assembly of TRIM5$\alpha$ during sedimentation. Human TRIM5$\alpha$ proteins were not used in these assays because they tended to aggregate and pellet, even in the absence of HIV-1 cores. TRIMCyp was also not used owing to low levels of residual cyclosporine A in the core preparations. Aliquots (5 µl) of the mixtures were mixed with 2X SDS-PAGE sample loading buffer ('Input' in *Figure 6*). The mixtures were layered onto a 30% (w/v) sucrose cushion (4 ml, prepared in binding buffer lacking TCEP) and subjected to centrifugation at 149,632 g (Beckman SW50.1 rotor) for 2.5 hr to separate free TRIM5$\alpha$ and unassembled CA proteins from capsid-bound TRIM5 proteins and pelletable cores. Following centrifugation, aliquots (45 µl) of supernatant (500 µl in total) were mixed with 4X SDS-PAGE sample loading buffer, and the pellets were resuspended in 25 µl 1X SDS-PAGE sample loading buffer. The TRIM5 and CA proteins in the input (2%), supernatant (2%) and pellet (30%) were separated by 12% SDS-PAGE. The integrated intensities of protein bands on the western blots were measured using the Odyssey software (LI-COR Bioscience). The molar ratios of TRIM5$\alpha_{AGMpyg}$ to CA in the pellets were estimated from standard curves constructed from known amounts of TRIM5$\alpha_{AGMpyg}$ and CA loaded on the same gel.

## Screening for TRIM5 decoration of HIV-1 cores

TRIM5$\alpha$-core complexes were prepared by incubating TRIM5$\alpha_{rh}$ with affinity-purified hyperstable HIV-1 cores in 50 mM HEPES (pH 8.0) or CHES (pH 9.0), 0.1 mM TCEP buffer at 4°C. Decoration conditions were surveyed over a range of TRIM5$\alpha_{rh}$ concentrations (0.25–2 µM), NaCl concentrations (50 or 200 mM), and incubation times (2–92 hr). Conditions that gave the best TRIM5$\alpha$ decoration on HIV-1 cores with minimal TRIM5$\alpha$ self-assemblies were determined by negative stain TEM imaging on a JEOL JEM-1400 Plus transmission electron microscope as described above. Clear decoration was difficult to discern in the majority of the TRIM5$\alpha$-incubated cores, likely because TRIM5$\alpha$ decorations were obscured by uranyl acetate staining of the underlying viral ribonucleoprotein complexes, and cores with clear external TRIM5$\alpha$ decoration patterns were observed on only ~5% of 907 core particles examined. Cores incubated in the absence of TRIM5$\alpha$ proteins and stained under the same conditions never showed equivalent decorations.

## ECT of TRIM5$\alpha$-decorated HIV-1 cores and tubes

### Cryo-grid preparation

TRIM5$\alpha$-decorated CA tubes were prepared by incubating 1.5 µM TRIM5$\alpha_{AGMpyg}$ with 695 µM wild type CA in 90 µl of assembly buffer (20 mM HEPES, 50 mM NaCl, pH 8.0) at 37°C for 2 hr, followed by a 2 hr incubation at room temperature. TRIM5$\alpha$-decorated HIV-1 cores were prepared by incubating 0.5 µM TRIM5$\alpha_{AGMpyg}$ with 0.6 µM CA equivalents of hyperstable HIV-1 cores in 75 µl of binding buffer (40 mM HEPES, 50 mM NaCl, 1 mM TCEP, pH 7.2) at 4°C for 1 hr, a condition that produced saturation binding in the co-sedimentation assay (data not shown). TRIM5$\alpha$-tube complexes, cores, or TRIM5$\alpha$-core complexes were mixed with BSA-coated colloidal gold particles (10 nm, SPI Supplies), which served as fiducials required for aligning the tilt stacked images. For cross-linked complexes, the pH of the samples was adjusted to 8.0 by adding 1/20[th] the sample volume of 1 M HEPES, pH 8.0, and the samples were incubated with 1 mM Sulfo-EGS (Pierce Biotechnology) at 23°C for 10 min in 88 µl total reaction volume. The cross-linking reaction was quenched by direct addition of 1 M Tris, pH 7.4, to a final concentration of 50 mM, followed by a 15 min incubation at 23°C. This treatment crosslinked ~87% and ~76% of the TRIM5$\alpha_{AGMpyg}$ and CA subunits of TRIM5-core complexes, respectively, as analyzed by SDS-PAGE and western blotting (data not shown). Samples (3.5 µl) were placed on the carbon side of freshly glow-discharged Quantifoil R2/2, 300 mesh holey carbon grids (SPI Supplies) for 1 min, thinned by automatic blotting using a Vitrobot Mark I (FEI) (-1.5 mm offset, 6–8 s, with filter papers from both sides at 80–85% relative humidity) and vitrified by plunge-freezing into liquid ethane. The cryo-grid was transferred to the microscope using a cryo-transfer holder.

## ECT

Images were collected using a 300 kV FEI G2 Polara transmission electron microscope equipped with an energy filter (slit width 20 eV; Gatan) and a 4k x 4k K2 Summit using the direct electron counting mode (Gatan). Pixels on the detector represented 0.26 nm (41,000x) at the specimen level. The tilt series were recorded from -60° to +60° with an increment of 1° and 4 μm underfocus. The cumulative dose of a tilt-series was 80–100 e-/Å$^2$. UCSF Tomo (*Zheng et al., 2007*) was used for automatic acquisition of the tilt series and 2D projection images. The tilt series was aligned and binned by 4 into 1k x 1k using the IMOD software package (*Kremer et al., 1996*), and 3D reconstructions were calculated using the simultaneous reconstruction technique (SIRT) implemented in the TOMO3D software package (*Agulleiro and Fernandez, 2011*), or weighted back projection using IMOD. Noise reduction was performed using the non-linear anisotropic diffusion (NAD) method in IMOD (*Kremer et al., 1996*), typically using a K value of 0.03–0.04 with 10 iterations.

## Segmentation and isosurface generation

Segmentation and isosurface rendering were performed in Amira (FEI). The outer boundary of the HIV-1 tube or core was first manually identified, and a material mask was generated inside the boundary. A second region of interest surrounding the tube or core that typically extended 9 nm from the exterior surface was generated (densities inside this region correspond to TRIM5α protein). The area inside the second region was segmented and an isosurface generated for the densities inside. Islands containing six voxels or fewer in 3D were deleted for the segmented cores, and four voxels or fewer were deleted for the co-assembled tube. The exterior layer of the CA protein within the HIV-1 tube or core of the first material was also segmented using a similar threshold value, and an isosurface was generated. Movie image sequences were generated in JPEG format in Amira (FEI) and converted into movies using QuickTime Player 7. Photoshop CS6 (Adobe) was then used to produce the final versions of the movies.

## Fitting the TRIM5 structural model to the cryoEM map

Crystal structures of the B-box 2 trimer (PDB 5EIA)(*Wagner et al., 2016*) and B-box 2/coiled-coil dimer (PDB 4TN3)(*Goldstone et al., 2014*) were fitted manually as separate units into the map using UCSF chimera (*Pettersen et al., 2004*). The trimer structures were initially placed at putative three-fold densities, and then iterative superpositions and manual adjustments were performed to optimize the overlap between the B-box 2 portions of the two source PDB files. No geometric optimization was performed, and the fitting should be treated as a simple proof of principle that the dimensions of the hexagon densities observed in the cryotomograms are compatible with dimensions and interactions observed in the crystal structures.

## Acknowledgements

Deep-etch electron microscopy was performed by Robyn Roth and John Heuser at the Laboratory of Electron Microscopy Sciences, Department of Cell Biology, Washington University School of Medicine. Some of the EM work was conducted at the Molecular Electron Microscopy Core Faciltiy at the University of Virginia, which is supported by the School of Medicine and built with NIH grant (G20-RR31199). We are grateful to Ruth Pumroy for contributing TRIM5 proteins used in our co-assembly assays, to Katherine Ferrell for the gift of His$_6$-Usp2 enzyme, and to members of our laboratories for helpful discussions and critical reading of the manuscript. This work was supported by funds from NIH NIGMS P50 082545 (to MY, BKG-P, GJJ and WIS).

## Additional information

### Competing interests

WIS: Reviewing editor, *eLife*. The other authors declare that no competing interests exist.

## Funding

| Funder | Grant reference number | Author |
|---|---|---|
| National Institutes of Health | NIGMS P50 082545 | Mark Yeager<br>Barbie K Ganser-Pornillos<br>Grant J Jensen<br>Wesley I Sundquist |

The funders had no role in study design, data collection and interpretation, or the decision to submit the work for publication.

## Author contributions

Y-LL, VC, BKG-P, Conception and design, Acquisition of data, Analysis and interpretation of data, Drafting or revising the article, Contributed unpublished essential data or reagents; SDC, CLW, DEC, Acquisition of data, Analysis and interpretation of data, Drafting or revising the article; KAD, Acquisition of data, Analysis and interpretation of data; OP, Conception and design, Acquisition of data, Analysis and interpretation of data, Drafting or revising the article; MY, GJJ, WIS, Conception and design, Analysis and interpretation of data, Drafting or revising the article

## Author ORCIDs

Viswanathan Chandrasekaran, http://orcid.org/0000-0002-0871-4740
Owen Pornillos, http://orcid.org/0000-0001-9056-5002
Grant J Jensen, http://orcid.org/0000-0002-7095-3507
Wesley I Sundquist, http://orcid.org/0000-0001-9988-6021

## Additional files

### Supplementary files

• Supplementary file 1. Plasmids and pure recombinant TRIM5 proteins used in this study. (A) List of plasmids used in this study. (B) TRIM5 constructs and their molecular weights.

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
