## [Decision Letter]

Congratulations: we are very pleased to inform you that your article, "Primate TRIM5 proteins form hexagonal nets on HIV-1 capsids", has been accepted for publication in *eLife*. The Reviewing Editor for your submission was Stephen Goff and the Senior Editor was Wenhui Li.

All three of our reviewers were enthused about the findings and the paper. They all summarized the salient points and raised only very minor points that can be readily addressed.

Reviewer #1 (General assessment and major comments (Required)):

The authors previously reported that TRIM5 forms a planar, hexagonal lattice and generated a model for how the TRIM5 subdomains are arranged. These experiments were limited in terms of detail and the use of a TRIM5 fusion protein (TRIM5-21R). Here electron cryotomograms (ECT) with full-length and SPRY-deleted TRIM5-21R confirmed the placement of the SPRY domain at the middle of the hexagon edges. The authors then developed a method for generating full-length, authentic TRIM5 proteins and confirmed hexameric lattice formation with 2 TRIM5 orthologues. 2D CA crystals were assembled and dimeric TRIM5 was added under conditions that preclude spontaneous lattice formation by TRIM5. CA-templated assembly was observed with 3 restricting TRIM5s but not with 2 non-restrictive TRIM5s. The purified TRIM5s co-sedimented with recombinant, crosslinked CA tubes. Negative stain and deep-etch EM demonstrated thin nets on the surface of the CA tubes when restrictive TRIM5s were added, but not with controls. The authors made virion cores by detergent extraction and density sedimentation gradient, expertly increasing the core yield with CA A14C/E45C/A92E, and enrichment on cyclophilin A columns. Cores were then incubated with TRIM5 protein and visualized by ECT. Hexagonal nets of TRIM5 were visualized and modeled. This expert body of work shows how restriction factor TRIM5 associates with HIV-1 cores. I have a few minor comments that might help the reader.

Reviewer #1 (Minor Comments):

1) Figure 1: The density difference map in 1E does not seem to match the difference of the representative images shown in the bottom middle panels of 1C and D. It looks like TRIM5-21R-_ΔSPRY_ retains density in the middle of the hexagon edges. Perhaps the authors could clarify this?

2) Concerning TRIM5 purification (subsection “Expression and purification of authentic primate TRIM5α and TRIMCyp proteins”) the authors should acknowledge that another group reported detailed methods for baculo expression and purification of high concentration, authentic TRIM5 from owl monkey (Pertel et al., 2011). Regarding the discussion at the end of the subsection “Reagent development”, this same group also used A14C/E45C stabilized CA cores to stimulate the Ubx E3 activity of purified TRIM5 protein.

3) Figure 3 shows the templated assembly of restrictive TRIM5 on 2D CA lattices. Images with a non-restrictive TRIM5 might help the reader visualize the differences.

4) Figure 4: The authors state that co-assembly of CA with TRIM5 was more efficient than if TRIM5 was added after CA was preformed into tubes (at the end of the subsection “TRIM5 binding to helical HIV-1 CA tubes”). This statement seems at odds with the histogram in Figure 4 which shows pretty much the same percentages of "Rings on preformed CA tubes" with "Rings on co-assembly".

5) In the subsection “Generation of hyperstable, disulfide-crosslinked HIV-1 core particle”: I don't understand the calculations stated in the text regarding Figure 5: “[…] a substantial fraction of the CA protein also concentrated at the density expected for native core particles (fractions 10-12, highlighted in pink in Figure 5, right panel). These "hyperstable" cores were reproducibly recovered in higher yields (0.8 ± 1%) than wild type cores (0.2 ± 0.1%)[…] “Fraction 10 looks to be 25%, not 0.8%?

Reviewer #2 (General assessment and major comments (Required)):

In their manuscript, Li and coworkers report the purification of milligram quantities of TRIM5 restriction factor proteins and structural analysis of complexes of some of them with purified HIV-1 cores and assemblies of recombinant HIV-1 CA protein. The authors also describe a new approach they developed to purify HIV-1 cores from virions. The study represents a significant advance because: (1) TRIM5 proteins have been notoriously difficult to purify, and (2) interactions of TRIM5α with native HIV-1 cores have not been described in the literature. Based on their structural results, the authors conclude that TRIM5 proteins form hexagonal "nets" around the viral capsid resembling those formed spontaneously by the restriction factors. Assuming the technique employed for purifying TRIM5 proteins is robust, this study should greatly facilitate future studies of TRIM5 structure and biochemistry.

I have no significant criticisms.

Reviewer #2 (Minor Comments):

Introduction, third paragraph: the term "viral core replication particle." I don't recall reading this term previously, and it seems like an odd combination of adjectives and nouns.

At the end of the subsection “Structure-based models for TRIM5-21R assembly”: “[…]thereby validating the model[…] “According to Merriam-Webster, the word "validate” implies establishing validity by authoritative affirmation or by factual proof. I suggest replacing this with "supporting."

Reviewer #3 (General assessment and major comments (Required)):

Li et al. report on a fairly comprehensive structural study focused on the nature of the interaction between TRIM5α and incoming retroviral capsid cores. The results to some extent confirm expectations based on highly artificial systems, and greatly extend these through extensive efforts to use bona fide, purified TRIM5α proteins together with capsid preps as close as possible to native structures. This is an important step forward – despite intense focus on TRIM5 and TRIM5-mediated restriction, standard approaches to understanding protein interactions have not been useful. As the authors point out, the methods may also be applicable to the study of the many other known capsid-interacting co-factors in the cell. Another strength of the study is that most results apply to two or more distinct TRIM5 homologs, confirming the generality of the findings. The model is also consistent with known interactions and with the observation that TRIM5-mediated restriction is saturable.

The results also begin to fill in/suggest aspects of mechanism that will certainly drive further work by both structural and cellular laboratories, and this will be a great strength of the paper. Interesting details emerging from the study include the likely intermolecular dimerization of RING domains upon assembly, which suggests a regulatory mechanism that among other things may control the E3 Ub ligase activity in the absence of a target or before sufficient assembly has occurred on the target, and the orientation of the RING domains on the exposed surface of the lattice. Results also indicate that the formation of a hexagonal lattice is a conserved mechanism, and open the door for structural studies to understand both the broad spectrum of retroviruses that can be recognized while also trying to understand how, at the same time, individual retroviruses can "escape" this interaction while maintaining integral capsid functions. It potentially may suggest how other TRIM proteins recognize their respective ligands.

Overall this is an important study and it should have a significant impact on the field.

---

## [Author Response]

Reviewer #1 (Minor Comments):

1) Figure 1: The density difference map in 1E does not seem to match the difference of the representative images shown in the bottom middle panels of 1C and D. It looks like TRIM5-21R-_ΔSPRY_ retains density in the middle of the hexagon edges. Perhaps the authors could clarify this?

An important difference between the TRIM5-21R and TRIM5-21R_ΔSPRY_ assemblies is that in the absence of the SPRY domains, multiple hexagonal lattices stacked on "top" of one another close together, laterally offset by a quarter, half, or three-quarters the distance across a hexagon. Thus in projections through 3-D sub-tomogram averages, like those shown in the bottom middle panels of 1C and D, all of these other offset lattices appear, but less prominently than the main lattice. Due to the special pattern of the offsets (one quarter, half, and three-quarters across) and the hexagonal geometry, their projections all intersect at the center of the arms of the main lattice, causing that position to appear especially dark and large. Furthermore, the four-helix bundle of both TRIM5-21R and TRIM5-21R_ΔSPRY_ also contributed to the observed densities in the center of the hexagon edges. Figure 1 shows, however, a thin slice through the 3-D difference map containing only the main hexagonal lattice, revealing the position of the SPRY domain without interference from the lattices above and below. These clarifications have now all been added to the Materials Section (subsection “TRIM5-21R electron cryo-tomography (ECT)”, fourth paragraph).

2) Concerning TRIM5 purification (subsection “Expression and purification of authentic primate TRIM5α and TRIMCyp proteins”) the authors should acknowledge that another group reported detailed methods for baculo expression and purification of high concentration, authentic TRIM5 from owl monkey (Pertel et al., 2011). Regarding the discussion at the end of the subsection “Reagent development”, this same group also used A14C/E45C stabilized CA cores to stimulate the Ubx E3 activity of purified TRIM5 protein.

The reviewer is correct, and we have now included both of those points (subsection “Expression and purification of authentic primate TRIM5α and TRIMCyp proteins”, first sentence and subsection “Reagent development”, last sentence).

3) Figure 3 shows the templated assembly of restrictive TRIM5 on 2D CA lattices. Images with a non-restrictive TRIM5 might help the reader visualize the differences.

We have added a new panel to Figure 3 with the requested image (Figure 3 in the revised manuscript).

4) Figure 4: The authors state that co-assembly of CA with TRIM5 was more efficient than if TRIM5 was added after CA was preformed into tubes (at the end of the subsection “TRIM5 binding to helical HIV-1 CA tubes”). This statement seems at odds with the histogram in Figure 4 which shows pretty much the same percentages of "Rings on preformed CA tubes" with "Rings on co-assembly".

There is some confusion here. Figure 4 shows the distribution of inter-ring/protrusion *spacings* in different types of TRIM5-CA co-assemblies, and makes the point that they are all similar. In each case, the binned distributions total 100%. However, the fraction of CA assemblies that are decorated with TRIM5 rings is not the same in each case. In the co-assembly reactions, essentially every CA tube was decorated with TRIM5 rings, as visualized by negative stained EM. In contrast, only about 1/3 of the preformed CA tubes were decorated with TRIM5 (with some variability in different preparations). Thus, our statement is correct that “co-assembly of CA with TRIM5 was more efficient than if TRIM5 was added after CA was preformed into tubes”.

5) In the subsection “Generation of hyperstable, disulfide-crosslinked HIV-1 core particle”: I don't understand the calculations stated in the text regarding Figure 5: “[…]a substantial fraction of the CA protein also concentrated at the density expected for native core particles (fractions 10-12, highlighted in pink in Figure 5, right panel). These "hyperstable" cores were reproducibly recovered in higher yields (0.8 ± 1%) than wild type cores (0.2 ± 0.1%)[…] “Fraction 10 looks to be 25%, not 0.8%?

As the reviewer notes, the integrated intensity in fractions 10-12 is roughly 25% (Figure 5), but the recovered yield was 0.8% (as reported). The difference reflects relatively inefficient recovery during the final purification step, in which fractions 10-12 were pooled, diluted with buffer, centrifuged, and the pelleted cores were resuspended in buffer and quantified.

Reviewer #2 (Minor Comments):

Introduction, third paragraph: the term "viral core replication particle." I don't recall reading this term previously, and it seems like an odd combination of adjectives and nouns.

We now use the term “viral core” (Introduction, third paragraph).

At the end of the subsection “Structure-based models for TRIM5-21R assembly”: “[…]thereby validating the model[…] “According to Merriam-Webster, the word "validate” implies establishing validity by authoritative affirmation or by factual proof. I suggest replacing this with "supporting."

We now use the term “supporting” (subsection “Structure-based models for TRIM5-21R assembly”, last sentence).